# PIMT is a novel and potent suppressor of endothelial activation

Chen Zhang[1], Zhi-Fu Guo[1], Wennan Liu[1], Kyosuke Kazama[1], Louis Hu[1], Xiaobo Sun[1], Lu Wang[2], Hyoungjoo Lee[3], Lin Lu[4], Xiao-Feng Yang[5], Ross Summer[1], Jianxin Sun[1]*

[1]Center for Translational Medicine, Thomas Jefferson University, Philadelphia, United States; [2]Department of Cell and Developmental Biology, Perelman School of Medicine, University of Pennsylvania, Philadelphia, United States; [3]Quantitative Proteomics Resource Center, University of Pennsylvania, Philadelphia, United States; [4]Department of Cardiovascular Medicine, Ruijin Hospital, Shanghai Jiao Tong University School of Medicine, Shanghai, China; [5]Center for Metabolic Disease Research, Lewis Katz School of Medicine, Temple University, Philadelphia, United States

**Abstract** Proinflammatory agonists provoke the expression of cell surface adhesion molecules on endothelium in order to facilitate leukocyte infiltration into tissues. Rigorous control over this process is important to prevent unwanted inflammation and organ damage. Protein L-isoaspartyl O-methyltransferase (PIMT) converts isoaspartyl residues to conventional methylated forms in cells undergoing stress-induced protein damage. The purpose of this study was to determine the role of PIMT in vascular homeostasis. PIMT is abundantly expressed in mouse lung endothelium and PIMT deficiency in mice exacerbated pulmonary inflammation and vascular leakage to LPS(lipopolysaccharide). Furthermore, we found that PIMT inhibited LPS-induced toll-like receptor signaling through its interaction with TNF receptor-associated factor 6 (TRAF6) and its ability to methylate asparagine residues in the coiled-coil domain. This interaction was found to inhibit TRAF6 oligomerization and autoubiquitination, which prevented NF-κB transactivation and subsequent expression of endothelial adhesion molecules. Separately, PIMT also suppressed ICAM-1 expression by inhibiting its N-glycosylation, causing effects on protein stability that ultimately translated into reduced EC(endothelial cell)-leukocyte interactions. Our study has identified PIMT as a novel and potent suppressor of endothelial activation. Taken together, these findings suggest that therapeutic targeting of PIMT may be effective in limiting organ injury in inflammatory vascular diseases.

*For correspondence:
jianxin.sun@jefferson.edu

Competing interest: The authors declare that no competing interests exist.

## Editor's evaluation

A multidisciplinary approach combining biochemical, cellular, and Mass-spec methods shows that PIMT regulates immune responses. PIMT exerts its anti-inflammatory function by inhibiting TRAF6 and ICAM-1. This unexpected function of PIMT in suppressing endothelial activation could open new doors in developing therapeutic strategies against vascular diseases.

## Introduction

Lung endothelium serves as a physical barrier to blood constituents and acts as a gate keeper of organ inflammation (*Galley and Webster, 2004*; *Pries and Kuebler, 2006*). Although largely quiescent in the healthy lungs, endothelial cells are an early participant in the pathogenesis of many inflammatory conditions, such as the acute respiratory distress syndrome (ARDS). For instance,

endothelial cells produce many of the factors that recruit leukocytes to the lung after injury and critically control immune cell entry through the expression of membrane adhesion molecules (*Maniatis et al., 2008*; *Millar et al., 2016*). Thus, tight control over endothelial responses is critical for limiting organ damage to most inflammatory insults, and interventions that attenuate activation of these cells have emerged as a new paradigm in the treatment of inflammatory diseases (*Orfanos et al., 2004*).

Canonical NF-κB (nuclear factor kappa enhancer binding protein) signaling plays a pivotal role in endothelial cell activation (*Baeuerle, 1998*). Binding of proinflammatory cytokines to receptors, such as members of interleukin-1 receptor (IL-1R)/toll-like receptor (TLR) superfamily, leads to the activation of the IKK (IκB [NF-κB inhibitor] kinase) complex via recruiting cytoplasmic adaptors, including TNF receptor-associated factors (TRAFs). The activation of the IKK complex causes phosphorylation of IκBα, which leads to polyubiquitination and subsequent proteasomal degradation. The liberated NF-κB dimer then translocates to the nucleus wherein it binds to specific DNA sequences to promote the expression of cytokines and endothelial adhesion molecules (*Dunne and O'Neill, 2003*; *Szmitko et al., 2003*). TRAF6 is a RING-domain cognate ubiquitin ligase (E3) associated with the intracellular domain of the IL-1R/TLR that mediates immune responses through lysine 63 (K63)-linked ubiquitination of diverse signaling intermediators (*Chen, 2005*; *Deng et al., 2000*). Negative regulators of TRAF6 have been shown to suppress cellular inflammation, suggesting clinical potential in the treatment of inflammatory diseases (*Lv et al., 2018*; *Wang et al., 2006*).

As another hallmark of the vascular inflammatory process, leukocyte-endothelial adhesion is mediated by interactions between endothelial adhesion molecules and cognate receptors on immune cells (*Szmitko et al., 2003*). Intercellular adhesion molecule-1 (ICAM-1; CD54) is an inducible transmembrane protein expressed on the surface of endothelial cells, with known roles in leukocyte adhesion and transmigration (*Springer, 1994*; *Staunton et al., 1988*). In addition, ligand-induced ICAM-1 cross-linking acts as a signal transducer to promote proinflammatory effects (*Lawson and Wolf, 2009*). ICAM-1 is heavily N-linked glycosylated on extracellular Ig-like domains, and previous studies have demonstrated that this post-translational modification (PTM) is indispensable for the proper conformation and biological function of ICAM-1 during inflammatory responses (*Chen et al., 2014b*). However, the molecular mechanisms underlying the regulation of ICAM-1 glycosylation remain largely unknown.

As a conserved protein repair enzyme, PIMT utilizes S-adenosyl methionine (SAM/AdoMet) to form methyl esters on carboxyl positions of isoaspartyl (isoAsp) residues. IsoAsp is generated from asparagine (Asn) deamidation or aspartate (Asp) dehydration via succinimide intermediate under physiological or stressed conditions and usually disrupts protein structure and/or function (*Geiger and Clarke, 1987*; *Reissner and Aswad, 2003*). The methylated intermediate then releases a methyl group to reform succinimide through spontaneous decomposition, which converts isoAsp to Asp in order to restore protein function (*Desrosiers and Fanélus, 2011*; *Mishra and Mahawar, 2019*; *Reissner and Aswad, 2003*). It is increasingly recognized that Asn deamidation is significantly engaged in diverse biological functions such as cell apoptosis and the control of metabolic responses under inflammatory and stress conditions (*Deverman et al., 2002*; *Zhao et al., 2020*). PIMT-mediated protein carboxyl methylation (O-methylation) was recently reported to be critically important for cells, indicating it may functionally rival other types of protein PTMs (*Sprung et al., 2008*). For instance, overexpression of PIMT has been shown to extend longevity of bacteria, fruit flies, and plants (*Chavous et al., 2001*; *Ogé et al., 2008*; *Pesingi et al., 2017*). PIMT deficient mice exhibit increased isoAsp residual deposition and early neurological death due to grand mal seizures (*Kim et al., 1997*). Furthermore, we and others have shown that PIMT plays an important role in angiogenesis and cardiomyocyte survival under oxidative conditions (*Ouanouki and Desrosiers, 2016*; *Yan et al., 2013*).

Here, we demonstrate that PIMT is a potent negative regulator of endothelial activation and vascular inflammation. Mechanistically, we show that PIMT acts by inhibiting LPS-induced NF-κB activation through methylation of TRAF6 in ECs and by physically interacting with ICAM-1 to impact N-glycosylation and its functional interactions with immune cells. In summary, our findings suggest that PIMT-mediated protein O-methylation plays a central role in controlling vascular inflammation in the lung.

# Results

## PIMT mitigates LPS-induced mouse pulmonary vascular inflammation and lung injury

Lung infection sustained by various pathogens, such as bacteria and the COVID-19, is the leading cause of ALI(acute lung injury)/ARDS, and its severity and morbidity are markedly increased in the elderly (*Akbar and Gilroy, 2020*; *Kang and Jung, 2020*). PIMT plays a role in protein repair by converting D-aspartyl and L-isoaspartyl residues resulting from spontaneous deamidation back to the normal L-aspartyl form (*Figure 1—figure supplement 1*). Despite its function as an anti-aging and stress protein, the role of PIMT in ALI/ARDS remains completely unknown. In this regard, we sought to investigate the pathological significance of PIMT in ALI induced by LPS. Since homozygous deletion of both *PIMT* alleles leads to fatal epileptic seizures at 30–60 days after birth (*Kim et al., 1997*), $Pimt^{+/-}$ mice and their wild-type (WT) counterparts were deployed in this study.

Following LPS injury, we found that susceptibility to ALI was significantly increased in $Pimt^{+/-}$ mice. This correlated with increased total protein levels and numbers of total leukocytes and neutrophils in isolated bronchoalveolar lavage fluid (BALF) from $Pimt^{+/-}$ mice compared to WT counterparts (*Figure 1A and B*). Furthermore, we found that BALF cytokine and chemokine levels for TNF alpha (TNF-α), interleukin-6 (IL-6) and C-C motif chemokine ligand 2 (Ccl2) were also significantly increased in $Pimt^{+/-}$ mice (*Figure 1C*). To further evaluate the role of PIMT in pulmonary inflammation, we examined the expression of adhesion molecules and inflammatory cytokines in lung tissues. Baseline expression of TNF-α, interleukin 1 beta (IL-1β), IL-6, and Ccl2 was indistinguishable between $Pimt^{+/-}$ and WT mice. However, after LPS instillation, mRNA levels for these factors were significantly increased in $Pimt^{+/-}$ mice compared to WT controls (*Figure 1D*).

Likewise, the expression of adhesion molecule ICAM-1 was also significantly increased in $Pimt^{+/-}$ mice, as determined by both quantitative real-time RT-PCR (qRT-PCR) (*Figure 1D*) and western blot (*Figure 1E*). Histological analysis displayed exaggerated lung injury in PIMT$^{+/-}$ mice, as manifested by the presence of increased alveolar hemorrhage, infiltration of inflammatory cells into distal air sacs and disruption of alveolar walls in response to LPS (*Figure 1F*). Immunofluorescent (IF) staining localized PIMT expression to the endothelium (*Figure 1G*), suggesting endothelial PIMT expression may be important for regulating lung inflammation.

## PIMT inhibits endothelial NF-κB transactivation

LPS has been known to stimulate the expression of inflammatory molecules in ECs via the TLR-NF-κB pathway (*Dauphinee and Karsan, 2006*). To define the role of PIMT in lung endothelial activation, we examined the effect of PIMT overexpression on NF-κB promoter activity at the transcriptional level. EA.hy926 endothelial cells were transiently transfected with a plasmid containing a heterologous promoter driven by NF-κB elements upstream of the luciferase gene (*Ahn et al., 1995*), and overexpression of PIMT markedly reduced the NF-κB driven luciferase activity under both basal and LPS-stimulated conditions. By contrast, ectopic expression of an inactive PIMT mutant, which has a deleted catalytic motif I (PIMT-Del; *Kagan and Clarke, 1994*), had limited effect on LPS-induced NF-κB activation (*Figure 2A and B*), suggesting the enzymatic activity is indispensable for inhibitory function. Meanwhile, knockdown of PIMT expression by lentivirus expressing *PIMT* shRNAs(small hairpin RNAs) significantly augmented LPS-stimulated NF-κB promoter-driven luciferase activity, as compared with the control shRNA (sh-Ctrl; *Figure 2C*). In human umbilical vein endothelial cells (HUVECs), PIMT depletion also augmented the expression of adhesion molecules and cytokines, including ICAM-1, vascular cell adhesion molecule 1 (VCAM1), TNF-α, IL-6, and Ccl2, as determined by qPCR, under both basal and LPS-stimulated conditions (*Figure 2D*), while adenovirus-mediated PIMT overexpression suppressed the expression of inflammatory molecules (*Figure 2—figure supplement 1*). Collectively, these results suggest PIMT is a novel negative regulator of NF-κB activation in ECs.

Canonical NF-κB pathway is transduced from cell surface receptors to nuclear events, and various effectors are implicated for this signaling cascade (*Lawrence, 2009*). To further identify components of NF-κB signaling regulated by PIMT, we examined the phosphorylation of IKKβ and IκBα. The decision to focus on these molecules was because they are considered major control points in the inflammatory cascade. Consistent with this, we found that IKKβ and IκBα phosphorylation levels were significantly attenuated by PIMT overexpression at different time points after LPS stimulation compared with control adenovirus (Ad-LacZ; *Figure 2E*). Furthermore, LPS-induced IKKβ and IκBα

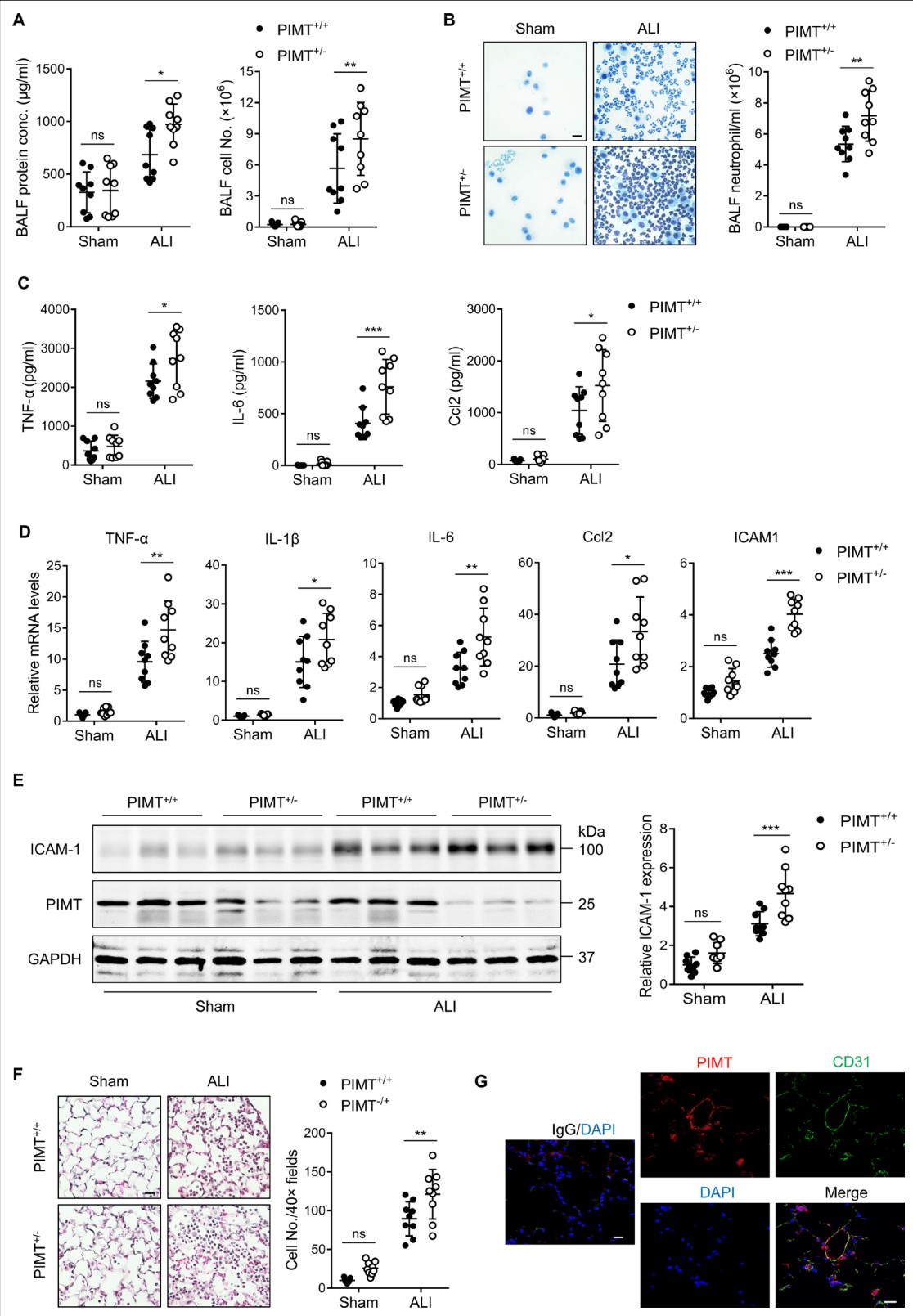

**Figure 1.** PIMT haploinsufficiency exacerbates LPS-induced mouse pulmonary vascular inflammation and lung injury. (**A**) Total cell counts and protein concentrations were determined in bronchoalveolar lavage fluid (BALF) of wild-type (WT; *Pimt*+/+) and *Pimt* hemizygous (*Pimt*+/−) mice 18 hr after intratracheal LPS (100 µg/100 µl, ALI) or PBS (100 µl, Sham) administration. (**B**) Diff-Quik stained smears of BALF (left) and neutrophil counts (right) were determined in WT and PIMT+/− mice 18 hr after intratracheal PBS or LPS administration. Bars, 20 µm. (**C**) Levels of TNF-α, IL-6, and Ccl2 in the BALF of

*Figure 1 continued on next page*

*Figure 1 continued*

WT and *Pimt*[+/–] mice were measured by enzyme-linked immunosorbent assay (ELISA) 18 hr after intratracheal PBS or LPS administration. (**D**) Expression of TNF-α, IL-1β, IL-6, Ccl2, and ICAM-1 mRNA extracted from lung tissues of WT and *Pimt*[+/–] mice with indicted treatments was determined by quantitative real-time RT-PCR (qRT-PCR). (**E**) Expression of ICAM-1 from lung tissues of WT and *Pimt*[+/–] mice with indicted treatments was determined by western blot and quantitated by densitometric analysis. (**F**) Hematoxylin and eosin staining of the lung sections of WT and *Pimt*[+/–] mice 18 hr after intratracheal PBS or LPS. Infiltrated immune cells were counted. Bars, 20 μm. (**G**) Representative immunofluorescent staining of PIMT in WT mouse lung sections. CD31 is shown as an endothelial marker. Nuclei were stained with DAPI (4′,6-diamidino-2-phenylindole), and IgG was used as a negative control. Bars, 20 μm. All data are expressed as mean ± SD, *p<0.05, **p<0.01, ***p<0.001, two-way ANOVA coupled with Tukey's post hoc test, and n=9 per group.

The online version of this article includes the following source data and figure supplement(s) for figure 1:

**Source data 1.** Expression of ICAM-1 from lung tissues of wild-type (WT) and PIMT[+/–] mice after LPS instillation.

**Figure supplement 1.** Catalytic process of PIMT-mediated isoaspartyl (isoAsp) methylation and protein repair cycle.

phosphorylation were further augmented in PIMT depleted cells as compared with the sh-Ctrl transduced cells (*Figure 2F*). Moreover, LPS-induced nuclear translocation of the p65 subunit of NF-κB, as determined by western blot and IF staining, was significantly reduced in PIMT overexpressing cells (*Figure 2G and H*). NF-κB DNA binding capacity, as detected by electrophoretic mobility-shift assay (EMSA), was significantly inhibited by overexpressing PIMT in ECs (*Figure 2I*). Together, these results indicate that PIMT inhibits LPS-induced endothelial NF-κB activation through acting on IKKβ kinase or its upstream regulators.

## PIMT interacts with TRAF6

TRAFs are critically involved in TNF receptors and TLR/IL-1 signaling pathways that lead to NF-κB activation (*Chen, 2005*). To define the regulatory targets of PIMT in TLR pathway, we performed a luciferase assay by co-transfection of NF-κB reporter plasmid together with expression vectors of candidate targets.

As shown in *Figure 3A*, overexpression of PIMT in ECs significantly inhibited the NF-κB activation induced by TRAF2 and TRAF6, but not by IKKβ-SS/EE, a constitutively active mutant of IKKβ (*Nottingham et al., 2014*). Furthermore, we examined the interaction of PIMT with TRAF2, TRAF6, and IKKβ by immunoprecipitation (IP) using cell lysates from HEK-293T cells exogenously expressing tagged proteins. Our co-IP demonstrated that PIMT was substantially enriched in the anti-TRAF6 immunocomplex, as compared with anti-TRAF2 or anti-IKKβ immunocomplexes (*Figure 3B*). The interaction of PIMT with TRAF6 was further confirmed by forward and reciprocal co-IP assays (*Figure 3C and D*). The endogenous interaction of TRAF6 with PIMT was confirmed by co-IP using an anti-PIMT antibody in both HUVECs and human lung microvascular endothelial cells (HLMVECs; *Figure 3E* and *Figure 3—figure supplement 1A, B*). Noteworthy, this endogenous interaction was augmented by LPS stimulation. In shRNA-mediated TRAF6 depleted cells, TRAF6 at an MW(molecular weight) of 60 kDa was absent in PIMT immunocomplexes, further confirming the specificity of PIMT interaction with TRAF6 in ECs. Given the fact that PIMT expression was marginally influenced by LPS in vivo or in cultured HLMVECs (*Figure 1E*; *Figure 3—figure supplement 1C*), these data strongly indicate that dynamic interaction of PIMT and TRAF6 may constitute an important negative feedback mechanism in resolving LPS inflammation.

TRAF6 is an ubiquitin E3 ligase which consists of a RING domain at its N-terminus, followed by zinc finger (ZnF) domains, a coiled-coil (CC) domain, and a TRAF-C domain at the C-terminus (*Chung et al., 2002*). To map the interacting domain(s) of PIMT in TRAF6, we constructed TRAF6 deletion mutants and determined interactions with PIMT by IP. As shown in *Figure 3F*, HA-tagged TRAF6 lacking the N-terminal RING dom ain and ZnF domains was efficiently pulled down by Flag-tagged PIMT, while further deletion of the CC domain comprehensively abolished the interaction. This observation was further confirmed in HEK-293T cells overexpressing Flag-tagged PIMT and Myc-tagged CC domain by IP (*Figure 3G*). Together, our results demonstrate that the CC domain of TRAF6 is responsible for the binding of PIMT in TRAF6.

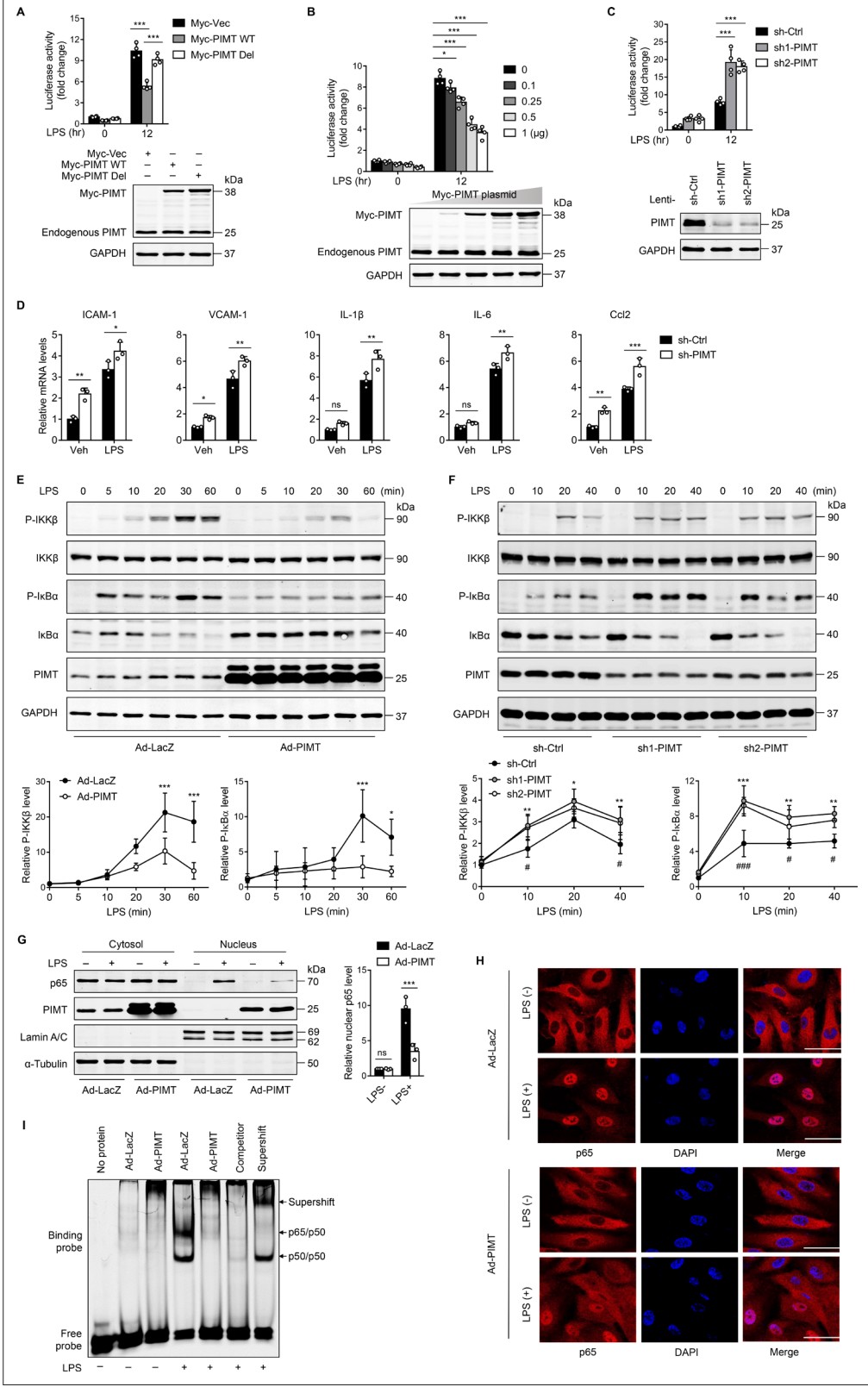

**Figure 2.** PIMT suppresses LPS-induced endothelial NF-$\kappa$B transactivation and proinflammatory effect. (**A**) EA.hy926 cells were transfected with NF-$\kappa$B-luciferase reporter together with either empty or Myc-tagged PIMT plasmids. 36 hr after transfection, cells were treated with either vehicle or LPS (1 µg/ml) for additional 12 hr and then harvested for the luciferase assay. Results are presented relative to Renilla activity (n=4). Endogenous and

*Figure 2 continued on next page*

*Figure 2 continued*

ectopic expression of PIMT were determined by western blot. Vec, pCS2–6×Myc empty vector; WT, wild-type PIMT; Del, catalytic defective PIMT mutant. (**B**) EA.hy926 cells were co-transfected with NF-κB luciferase reporter together with increased amounts of Myc-tagged PIMT plasmid. 36 hr after transfection, cells were stimulated with either vehicle or LPS (1 µg/ml) for 12 hr and then harvested for luciferase assay (n=4). Expression of PIMT was determined by western blot. (**C**) EA.hy926 cells were transduced with lentivirus expressing control shRNA (sh-Ctrl) or PIMT shRNAs (sh1-PIMT or sh2-PIMT) for 36 hr and then transfected with NF-κB-luciferase reporter plasmid for additional 36 hr. Luciferase activity was determined after LPS (1 µg/ml) treatment for 12 hr (n=4). PIMT levels were determined by western blot. (**D**) Human umbilical vein endothelial cells (HUVECs) were transduced with lentivirus expressing sh-Ctrl or PIMT shRNAs (sh1-PIMT or sh2-PIMT) for 60 hr. The mRNA expression of ICAM-1, VCAM1, IL-1β, IL-6, and Ccl2 was determined by quantitative real-time RT-PCR (qRT-PCR) after LPS (200 ng/ml) treatment for 12 hr (n=3). (**E**) HUVECs were transduced with Ad-LacZ or Ad-PIMT (moi = 10) for 48 hr and then stimulated with LPS (1 µg/ml) for indicated time points. Expression of PIMT and phosphorylation of the indicated proteins in toll-like receptor (TLR)-mediated NF-κB pathway was detected by western blot (above) and quantitated by densitometric analysis (n=3). (**F**), HUVECs were transduced with lentivirus expressing sh-Ctrl or PIMT shRNAs (sh1-PIMT or sh2-PIMT) for 72 hr and stimulated with LPS (200 ng/ml) for indicated time points. Expression and phosphorylation of indicated proteins were determined by western blot and quantitated by densitometric analysis (n=3). *p<0.05, **p<0.01, and ***p<0.001 (sh-Ctrl vs. sh1-PIMT). #p<0.05 and ###p<0.001 (sh-Ctrl vs. sh2-PIMT). Two-way ANOVA coupled with Tukey's post hoc test. (**G**) HUVECs were transduced with Ad-LacZ or Ad-PIMT (moi = 10) for 48 hr and then treated with LPS (1 µg/ml) for 30 min. Levels of NF-κB p65 in the cytoplasmic and nuclear fractions were determined by western blot. Lamin A/C and α-Tubulin were used as nuclear and cytoplasmic markers. The representative images (left) and quantification (right, n=3) were shown. (**H**) HUVECs were transduced with Ad-LacZ or Ad-PIMT (moi = 10) for 48 hr, followed by LPS (1 µg/ml) stimulation for 30 min. Localization of p65 was determined by immunofluorescent staining. Nuclei were stained with DAPI. Scale bars, 20 µm. (**I**) HUVECs were transduced with Ad-LacZ or Ad-PIMT (moi = 10) for 48 hr and then stimulated with LPS (1 µg/ml) for 30 min. Nuclear fraction was extracted, and NF-κB DNA-binding activity was determined by electrophoretic mobility-shift assay (EMSA). Competitor was 50-fold unlabeled probe. Supershift was incubated with anti-p65 antibody. All data are representative of mean ± SD, *p<0.05, **p<0.01, ***p<0.001, and two-way ANOVA coupled with Tukey's post hoc test.

The online version of this article includes the following source data and figure supplement(s) for figure 2:

**Source data 1.** Endogenous and ectopic expression of PIMT in EA.hy926 cells.

**Source data 2.** Expression of PIMT of PIMT in EA.hy926 cells transfected with Myc-PIMT plasmid.

**Source data 3.** Effect of PIMT shRNA on PIMT expression.

**Source data 4.** Effect of PIMT expression on NF-κB activation.

**Source data 5.** Effect of PIMT knockdown on NF-κB activation.

**Source data 6.** Effect of PIMT overexpression on NF-κB p65 cellular localization.

**Figure supplement 1.** PIMT suppresses LPS-induced expression of inflammatory molecules in ECs.

## PIMT inhibits TRAF6 function through methylation of Asn at position 350

The CC domain of TRAF6 has been shown to mediate TRAF6 oligomerization and its interaction with ubiquitin-conjugating enzymes, which is essential for the subsequent autoubiquitination that leads to NF-κB activation (*Wang et al., 2001*; *Wang et al., 2006*). Here, we confirmed that Flag-TRAF6 precipitated HA-TRAF6 in co-transfected HEK-293T cells (*Figure 4A*). However, this interaction was markedly inhibited by PIMT-WT, but not by inactive PIMT-Del. Similarly, the polyubiquitination of TRAF6 was exclusively inhibited by PIMT-WT (*Figure 4B*), suggesting that enzymatic activity of PIMT is required for the inhibition of TRAF6 function. Transforming growth factor-β (TGF-β)-activated kinase 1 (TAK1) is a substrate of TRAF6, and its activation has been implicated in LPS-mediated TLR signaling (*Irie et al., 2000*). Likewise, we found that both TAK1 ubiquitination and phosphorylation were inhibited by active PIMT but not by its inactive mutant (*Figure 4—figure supplement 1A, B*).

Asp (D) or Asn (N) following by small or flexible side chain amino acids such as glycine, histidine, or serine residing on protein surface structures is inclined to undergo dehydration or deamidation yielding isoAsp, which could be methylated by PIMT (*Geiger and Clarke, 1987*; *Reissner and Aswad, 2003*). To determine whether PIMT methylates TRAF6 under physiological conditions, the C-terminal domain of TRAF6 (TRAF6-ΔN) was immunoprecipited from either PIMT expressing or PIMT depleting HEK-293T

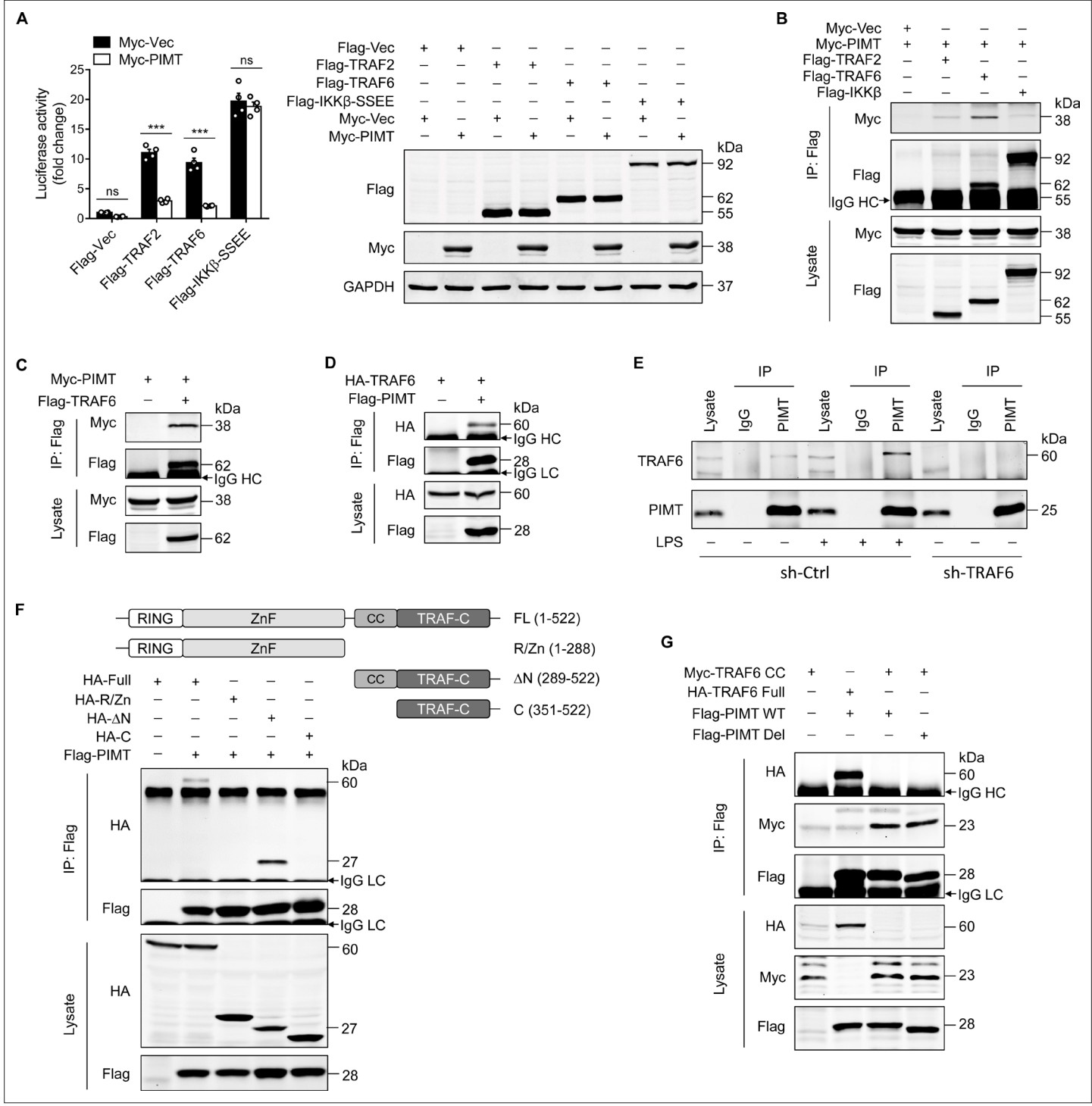

**Figure 3.** PIMT functionally interacts with TRAF6. (**A**) EA.hy926 cells were transfected with NF-κB luciferase reporter together with indicated plasmids. Cells were harvested 48 hr later to determine luciferase activity (n=4, ***p<0.001, and two-way ANOVA coupled with Tukey's post hoc test. Expression of transfected proteins was detected by western blot. (**B**) HEK-293T cells were transfected with a combination of indicated plasmids. 48 hr after transfection, immunoprecipitation was performed using anti-Flag antibody, followed by western blot to detect protein interaction. (**C**) HEK-293T cells were transfected with Flag-TRAF6 and Myc-PIMT expression vectors. 48 hr after transfection, immunoprecipitation was performed using anti-Flag antibody, followed by western blot to detect the interaction of PIMT with TRAF6. (**D**) HEK-293T cells were transfected with HA-TRAF6 and Flag-PIMT vectors. 48 hr after transfection, immunoprecipitation was performed using anti-HA antibody, followed by western blot to detect protein interaction. (**E**) Human umbilical vein endothelial cells (HUVECs) were transduced with lentivirus expressing the control shRNA (sh-Ctrl) or TRAF6 shRNA (sh-TRAF6) for 72 hr, followed by stimulation with either vehicle or LPS (1 μg/ml) for 30 min. Cell lysates were then collected for immunoprecipitation using anti-PIMT

*Figure 3 continued on next page*

*Figure 3 continued*

antibody, followed by western blot to detect the interaction of TRAF6 with PIMT. (**F**) HEK-293T cells were transfected with Flag-PIMT together with HA-tagged full length TRAF6 or its deletion mutants as indicated. 48 hr after transfection, immunoprecipitation was performed using anti-Flag antibody, followed by western blot to detect the interaction of PIMT with wild-type (WT) or truncated TRAF6 mutants. (**G**) HEK-293T cells were transfected with PIMT or its mutants together with tagged TRAF6 or its coiled-coil (CC) domain. Immunoprecipitation was performed using anti-Flag antibody, followed by western blot to detect the interaction of indicated proteins.

The online version of this article includes the following source data and figure supplement(s) for figure 3:

**Source data 1.** Expression of indicated proteins in transfected EA.hy926 cells.

**Source data 2.** Interaction of PIMT with TRAFs.

**Source data 3.** Interaction of PIMT with TRAF6.

**Source data 4.** Interaction of PIMT with TRAF6.

**Source data 5.** Interaction of PIMT with TRAF6 in human umbilical vein endothelial cells (HUVECs).

**Source data 6.** Interaction of PIMT with TRAF6 mutants.

**Source data 7.** Interaction of PIMT with TRAF6 coiled-coil (CC) domain.

**Figure supplement 1.** Endogenous interaction of PIMT with TRAF6 in ECs.

**Figure supplement 1—source data 1.** Interaction of PIMT with TRAF6 after LPS stimulation in human lung microvascular endothelial cells (HLMVECs).

**Figure supplement 1—source data 2.** Interaction of TRAF6 with PIMT in human umbilical vein endothelial cells (HUVECs).

**Figure supplement 1—source data 3.** Effects of LPS on ICAM-1 and PIMT expression.

cells (*Figure 4C*) and then subjected to nano liquid chromatography and mass spectrometry (nano LC-MS/MS) analysis to identify potential methylation sites. Our results demonstrated that D339 and N350, located in the CC domain (residues 292–350) of TRAF6, were methylated in PIMT expressing cells (*Figure 4D*, *Figure 4—figure supplement 1D E*). As expected, ~50% knockdown of PIMT would not completely abolish all of these methylations. Both PIMT-modified and unmodified N350 are present in PIMT depleted cells (*Figure 4—figure supplement 1F*). Interestingly, in addition to methylation, N350 deamination was abundantly detected in both PIMT expressing and knockdown cells, suggesting that the N350 deamination is naturally present regardless of the level of PIMT in cells. Since N350 is followed by a glycine, which is a typical 'hot spot' for deamidation and subsequent O-methylation by PIMT (*Reissner and Aswad, 2003*), we speculated that deamination of N350 can be further methylated in PIMT expressing cells.

D339 and N350 sites are conserved among species (*Figure 4E*). To determine their functional significance, we generated two methylation-dead TRAF6 mutants (D339A and N350A) and found that overexpression of D339A and N350A mutants in EA.hy926 cells significantly activated NF-κB to the similar extent as WT TRAF6 (*Figure 4F*). Importantly, co-overexpression of PIMT markedly inhibited both WT TRAF6- and D339A mutant-induced NF-κB transactivation but minimally affected N350A-induced NF-κB transactivation, suggesting that N350 O-methylation is involved in the PIMT-mediated suppression of NF-κB activation. Furthermore, N350A mutation also specifically attenuated PIMT-mediated inhibition of TRAF6 oligomerization and autoubiquitination (*Figure 4G and H*). Together, these findings suggest that O-methylation of TRAF6 at N350 is critically involved in the regulation of TRAF6 function and subsequent NF-κB activation by PIMT.

## PIMT impedes ICAM-1 expression and glycosylation in ECs

Activation of NF-κB has been implicated in the cytokine-induced expression of adhesion molecules, such as VCAM-1 and ICAM-1, which mediate the adhesion of monocytes to inflamed ECs (*Pober and Sessa, 2007*). Here, we found that overexpression of PIMT not only attenuated TNF-α-stimulated expression of ICAM-1 and VCAM-1 in a dose-dependent manner but also induced a surprise shift of protein bands high molecular weight (HMW) to lower molecular weight (LMW) forms (*Figure 5A*), which we speculate represents a partially glycosylated form of protein. Similar results were observed in LPS- and IL-1β-treated HUVECs (*Figure 5B*) and HLMVECs (*Figure 5—figure supplement 1A*), illustrating the broad relevance of our findings in endothelial biology. In attempt to determine whether PIMT impacts the global glycosylation pattern in ECs, we performed western blot using biotinylated lectin *Phaseolus vulgaris* leucoagglutinin (PHA-L) coupled with fluorescence labeled streptavidin staining of HUVEC lysates. No significant disparity was observed in Ad-LacZ- and Ad-PIMT-transduced ECs (*Figure 5—figure supplement 1B*). Additionally, we found PIMT knockdown by lentiviral shRNAs efficiently increased ICAM-1 protein levels in ECs, as

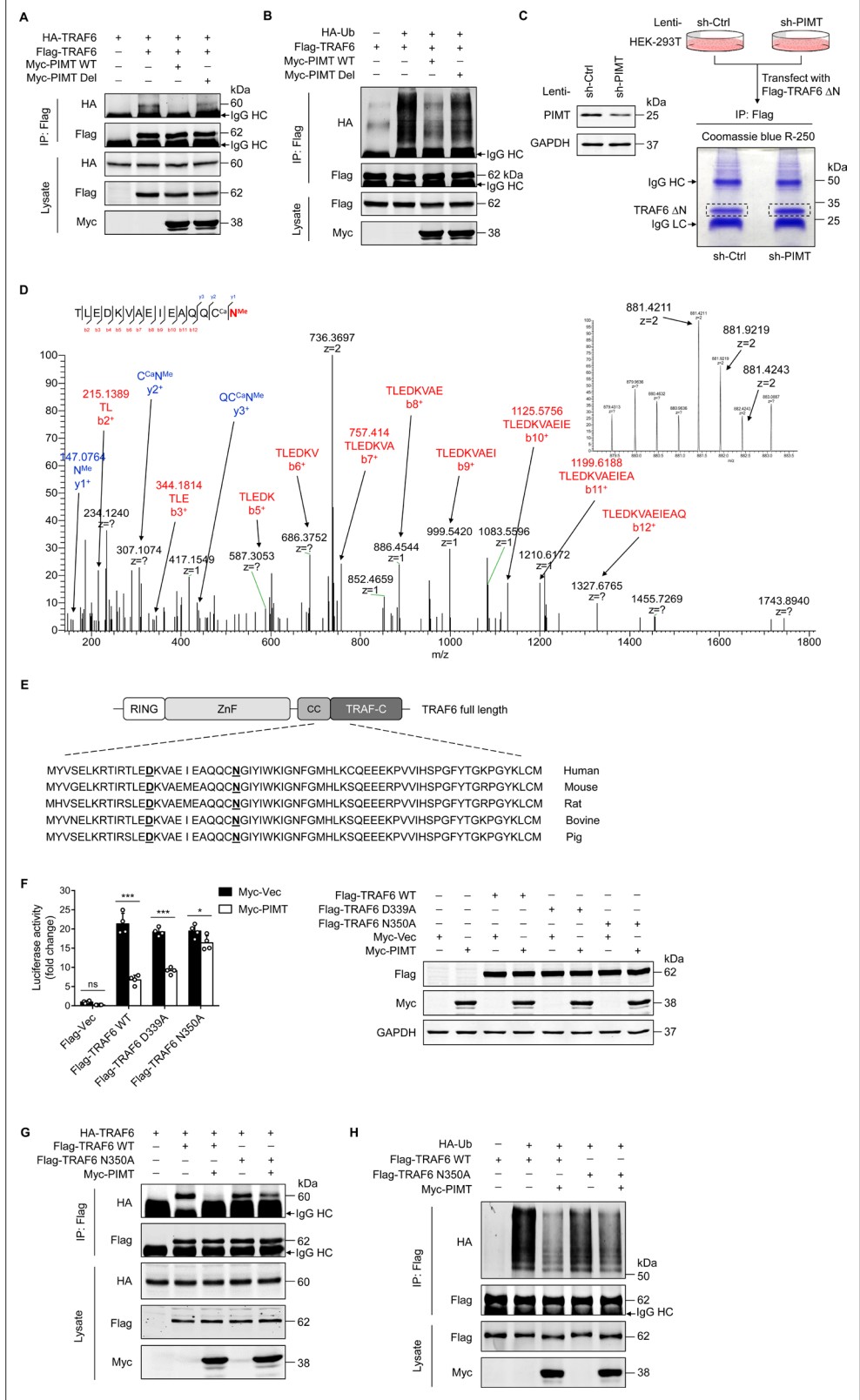

**Figure 4.** PIMT suppresses TRAF6 function through methylation of TRAF6 at N350. (**A**) HEK-293T cells were transfected with TRAF6 and PIMT constructs. Immunoprecipitation was performed 48 hr after transfection using anti-Flag antibody, followed by western blot using anti-HA antibody to detect TRAF6 oligomerization. (**B**) HEK-293T cells were transfected with HA-ubiquitin (HA-Ub) and Flag-TRAF6 in the presence of Myc-tagged wild-type

*Figure 4 continued on next page*

*Figure 4 continued*

(WT) PIMT and its enzymatic inactive mutant (PIMT Del). 48 hr after transfection, cells were collected for anti-Flag immunoprecipitation. The autoubiquitination of TRAF6 was detected by western blot using anti-HA antibody. (**C**) Left, western blot showing expression of PIMT in control or PIMT stable depleted HEK-293T cells. Right, work flow of in gel preparation of TRAF6 ΔN (shown in *Figure 3H*) for mass spectrometric analysis. (**D**) Liquid chromatography and mass spectrometry (LC-MS/MS) analysis of asparagine (Asn)/aspartate (Asp) modification in TARF6 ΔN domain. Mass tolerance window of the fragment ions is set within 0.02 Da. Ca, carbamidomethyl. Me, methyl. Left, manually assigned MS/MS spectra of TLEDKVAEIEAQQC (carbamidomethyl) N (methyl). Right, corresponding two charged precursor ion peaks, the m/z of the monoisotopic peak (m/z 841.4211, z=2) was found to be 1.4 ppm apart from the theoretical m/z (881.4198, z=2). (**E**) Alignment of TRAF6 protein sequences from different species. Identified methylated Asp and Asn residues are underlined. (**F**) EA.hy926 cells were transfected with NF-$\kappa$B reporter plasmid together with WT TRAF6 or TRAF6 mutants (D339A and N350A). 48 hr after transfection, cell were collected for luciferase assay (n=4), *p<0.05, ***p<0.001, and two-way ANOVA coupled with Tukey's post hoc test. Expression of transfected proteins were determined by western blot. (**G**) HEK-293T cells were transfected with HA-TRAF6 and indicated Flag-tagged TRAF6 constructs together Myc-PIMT plasmid. 48 hr after transfection, immunoprecipitation was performed using anti-Flag antibody, followed by western blot using anti-HA antibody to detect TRAF6 oligomerization. (**H**) HEK-293T cells were transfected with Flag tagged WT or mutant TRAF6 together with HA-Ub and Myc-PIMT plasmids. 48 hr after transfection, TRAF6 was immunoprecipitated by anti-Flag antibody, and autoubiquitination was then detected by western blot using anti-HA antibody.

The online version of this article includes the following source data and figure supplement(s) for figure 4:

**Source data 1.** Effect of PIMT on TRAF6 oligomerization.

**Source data 2.** Effect of PIMT on TRAF6 autoubiquitination.

**Source data 3.** Effect of lentivirus bearing PIMT shRNA on PIMT expression.

**Source data 4.** Expression of PIMT and TRAF6 in transfected EA.hy926 cells.

**Source data 5.** Role of TRAF6 N350 in TRAF6 oligomerization.

**Source data 6.** Role of TRAF6 N350 in TRAF6 autoubiquitination.

**Figure supplement 1.** PIMT suppresses TAK1 ubiquitination and phosphorylation.

**Figure supplement 1—source data 1.** Effect of PIMT on TAK1 ubiquitination.

**Figure supplement 1—source data 2.** Effect of PIMT on TAK1 phosphorylation.

determined by western blot (*Figure 5C*) and flow cytometry (*Figure 5—figure supplement 1C*), and the adhesion of monocytes to activated ECs (*Figure 5—figure supplement 1D*). Since ICAM-1 is essentially involved in both adhesion and extravasation of leukocytes during inflammation, it was chosen to further define the molecular mechanism of how ICAM-1 glycosylation was regulated by PIMT1. We used moderate multiplicity of infection (moi = 5) of Ad-PIMT to induce both HMW and LMW ICAM-1 and then performed co-IP. Our results showed that PIMT interacts only with LMW ICAM-1 (~70 kDa) but not with HMW ICAM-1 (~110 kDa; *Figure 5D and E*; *Figure 5—figure supplement 1E*), suggesting this interaction impairs the processing of ICAM-1 glycosylation in ECs.

ICAM-1 possesses a C-terminal cytoplasmic domain, a single transmembrane domain, and an extracellular domain containing eight N-glycosylation sites (*Staunton et al., 1988*). N-linked glycans are covalently attached to the amide side chain of Asn residues that resides in N-x-S/T consensus sequence (x represents any amino acid except proline). N-glycosylation impairment could induce major molecular weight shift (*Mellquist et al., 1998*). To determine whether PIMT affects the glycosylation of ICAM-1 at specific Asn sites, ICAM-1 was IP from the control and PIMT overexpressing ECs (*Figure 5F*) and then subjected to LC-MS/MS analysis to characterize the status of ICAM-1 N-glycosylation. Eight Asn residuals have been described as N-glyco sites in ICAM-1 in previous studies (*Casasnovas et al., 1998*; *Shimaoka et al., 2003*; *Yang et al., 2004*), while in PIMT overexpression samples, four of them were identified to be switched to Asp residues, which are incapable of linking N-glycans (*Figure 5G*; Figure S3C-E). Interestingly, all these sites were positioned in the antigen-binding regions of ICAM-1 extracellular Ig domains (*Figure 5H*). Together, these results suggest that PIMT prevents N-linked glycosylation of ICAM-1 through its enzymatic conversion of Asn to Asp at specific Asn sites.

## PIMT inhibits the function of ICAM-1 in endothelial activation

The canonical function of endothelial ICAM-1 and other adhesion molecules is involved in leukocyte arrest and extravasation (*Springer, 1994*). To determine whether PIMT affects ICAM-1 function in ECs, we firstly

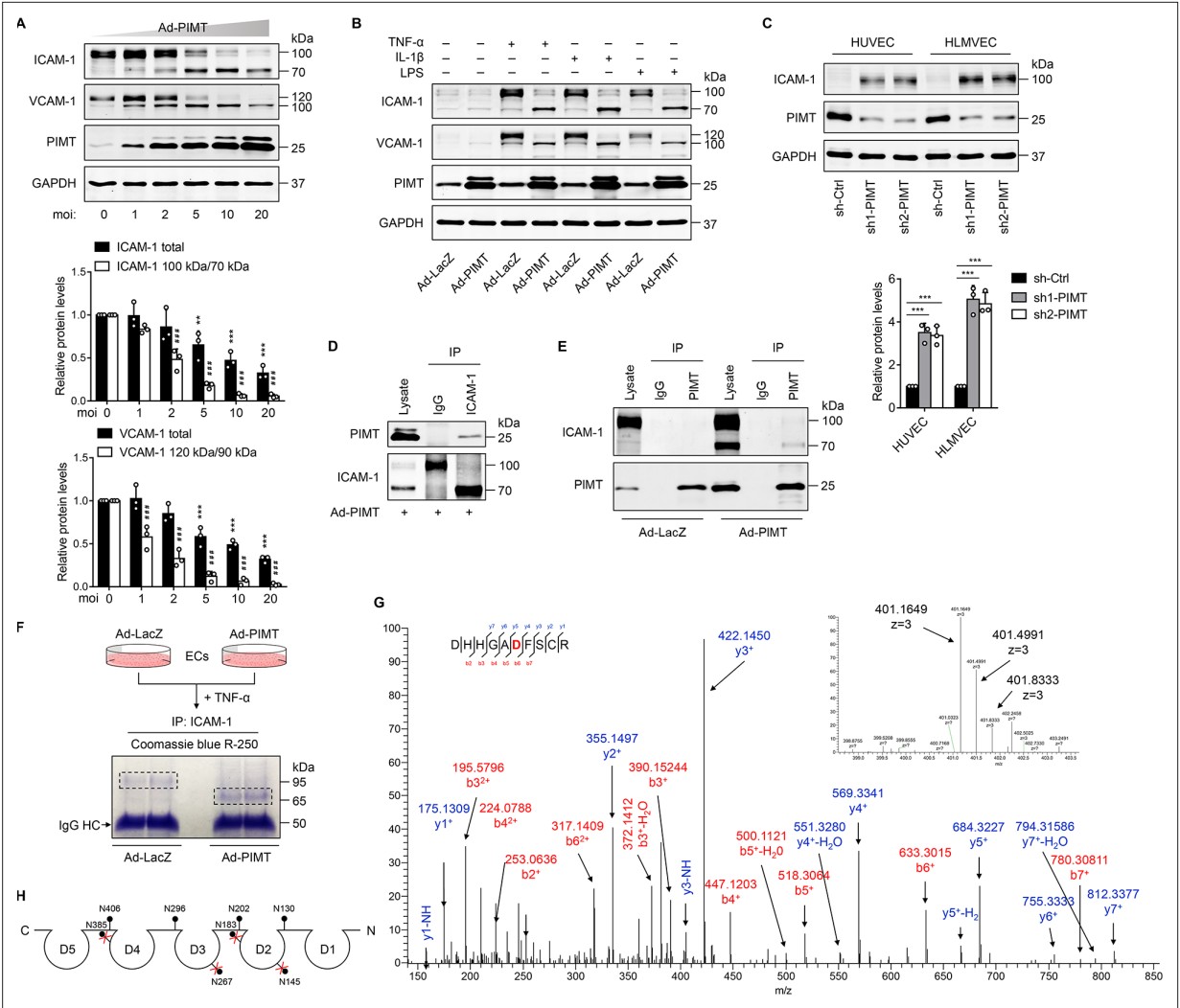

**Figure 5.** PIMT attenuates ICAM-1 expression and N-linked glycosylation in ECs. (**A**) Human umbilical vein endothelial cells (HUVECs) were transduced with increasing dosages of Ad-PIMT for 48 hr and then stimulated with TNF-α (10 ng/ml) for 12 hr. Expression of ICAM-1 and VCAM-1 was determined by western blot. Total protein levels of ICAM-1 and VCAM-1 and the ratio of high molecular weight (HMW) bands to lower molecular weight (LMW) bands were quantitated at different mois (normalized to GAPDH), n=3. **p<0.01, ***p<0.001, compared with the corresponding ICAM-1 or VCAM1 expression at moi of 0. ###p<0.001, compared with the corresponding the ratio of HMW bands to LMW bands of ICAM-1 or VCAM-1 at moi of 0, two-way ANOVA coupled with Tukey's post hoc test. (**B**) HUVECs transduced with indicated adenoviruses were exposed to TNF-α (10 ng/ml), IL-1β (20 ng/ml), or LPS (1 μg/ml) for 12 hr, and expression of ICAM-1 and VCAM-1 was detected by western blot. (**C**) HUVECs and human lung microvascular endothelial cells (HLMVECs) were transfected with lentivirus expressing control shRNA (sh-Ctrl) or PIMT RNAs (sh1-PIMT or sh2-PIMT) for 72 hr, and expression of ICAM-1 was detected by western blot and quantitated by densitometric analysis. ***p<0.001, one-way ANOVA coupled with Tukey's post hoc test, n=3. (**D**) HUVECs were transduced with Ad-PIMT (moi = 10) for 48 hr and then treated with TNF-α (10 ng/ml) for 12 hr. Immunoprecipitation was performed using anti-ICAM-1 antibody, followed by western blot to detect the interaction of ICAM-1 with PIMT. (**E**) HUVECs were transduced with Ad-PIMT (moi = 5) for 48 hr and then treated with TNF-α (10 ng/ml) for 12 hr. Immunoprecipitation was performed using anti-PIMT antibody, followed by western blot to detect the interaction of PIMT with ICAM-1. (**F**) Workflow in gel preparation of ICAM-1 from ECs for liquid chromatography and mass spectrometry (LC-MS/MS) analysis. Coomassie Blue R-250 staining showed LMW ICAM-1 immunoprecipitated from HUVECs transduced with Ad-PIMT. (**G**) Precursor ion peaks corresponding to two charged spectra, the m/z of the monoisotopic peak was exactly matched to the theoretical m/z (401.1649, z=3; above). Manually assigned MS/MS spectra of ICAM-1 peptide from PIMT overexpressed HUVECs (below). One representative spectrum is shown to exemplify the N-glycosite change. The asparagine (Asn) site was detected as aspartate (Asp). (**H**) Schematic demonstration of the N-glycosites of ICAM-1 (shown as lollipop patterns). Identified deamidated residuals were marked with red crosses.

The online version of this article includes the following source data and figure supplement(s) for figure 5:

**Source data 1.** Effect of PIMT on ICAM-1 and VCAM-1 expression.

**Source data 2.** Effect of PIMT on ICAM-1 and VCAM-1 expression in response to different inflammatory stimuli.

**Source data 3.** Effect of PIMT knockdown on ICAM-1 expression.

*Figure 5 continued on next page*

*Figure 5 continued*

**Source data 4.** Interaction of PIMT with ICAM-1.

**Source data 5.** Interaction of PIMT with low molecular weight ICAM-1.

**Figure supplement 1.** PIMT attenuates ICAM-1 expression in ECs.

**Figure supplement 1—source data 1.** Effect of PIMT on ICAM-1 expression in ECs.

**Figure supplement 1—source data 2.** Interaction of PIMT with lower molecular weight (LMW) ICAM-1 in ECs.

**Figure supplement 2.** PIMT affects N-linked glycosylation of ICAM-1 at specific Asn sites in ECs.

examined the membrane location of ICAM-1 by flow cytometry and found that TNF-α significantly increased cell surface expression of ICAM-1, which was markedly attenuated by ectopic expression of PIMT in ECs (*Figure 6A*). Furthermore, cell surface biotinylation assay revealed that both HMW and LMW forms of ICAM-1 were localized to the cell surface (*Figure 6B*), suggesting that glycosylation deficient ICAM-1 is able to escape the surveillance of endoplasmic reticulum quality control (ERQC) and localize to the cell membrane (*Jørgensen et al., 2003*). Notable, cellular fractionation revealed that LMW ICAM-1 was absent from the cytoskeleton (*Figure 6C*).

Next, we examined protein stability and found LMW ICAM-1 is stably preserved in HUVECs, while the rate of degradation of LMW ICAM-1 was markedly increased (*Figure 6D*). Monocyte adhesion to ECs is an important event in the initiation of proinflammatory response. As shown in *Figure 6E*, TNF-α stimulation substantially increased THP1 cell adhesion to ECs, which was markedly inhibited by overexpression of PIMT. Cross-linking of ICAM-1 on the surface of different cell types has previously been shown to cause an increase in cellular activation within the cytoplasm, which requires the interplay of ICAM-1 with the cytoskeleton (*Lawson et al., 1999*). To determine whether PIMT affects ICAM-1-mediated cell signaling events in ECs, we ligated cell surface ICAM-1 using a monoclonal antibody in HUVECs, followed by cross-linking with a secondary antibody and examined ICAM-1 inside-out signaling (*Lawson and Wolf, 2009*). As displayed in *Figure 6F*, phosphorylation of ERK1/2 was efficiently induced by specific antibody ligation but not by control IgG, and this signal was suppressed by overexpression of PIMT. Pre-treatment with tunicamycin (TM), which blocks the first step of N-glycan transfer in glycoprotein synthesis, also significantly attenuated ICAM-1 crosslinking aroused signaling. Together, these results suggest that PIMT functions as a negative regulator of the cytokine-induced inflammatory responses in vascular ECs, at least in part, through inhibiting expression and glycosylation of adhesion molecules in vascular ECs.

## Discussion

PIMT (encoded by Protein-L-Isoaspartate [D-Aspartate] O-Methyltransferase [*Pcmt1*] gene) is a protein carboxyl methyltransferase that specifically recognizes structurally altered carboxylic acids at aspartyl residues, and its role in monitoring protein homeostasis has been defined as 'housekeeping' function in a wide variety of cells and tissues (*Desrosiers and Fanélus, 2011*; *Mishra and Mahawar, 2019*; *Reissner and Aswad, 2003*). In this study, we identified a novel function of PIMT in suppressing endothelial activation and acute inflammatory responses. Specifically, we found that PIMT attenuates LPS-stimulated endothelial activation and acute lung injury by inhibiting NF-κB signaling through methylation of TRAF6 and inhibition of N-glycosylation of the cell surface adhesion molecule ICAM-1.

TRAF6 plays key roles in eliciting IL-1R/TLR-mediated inflammatory signaling (*Akira and Takeda, 2004*; *Chen, 2005*; *Deng et al., 2000*). TRAF6 consists of an N-terminal RING domain, followed by four ZnF domains, a CC domain, and a TRAF-C domain at the C-terminus (*Chung et al., 2002*). The RING domain, in association with ZnF domains, interacts with Ub-conjugating enzymes (E2) for Ub transfer (*Yin et al., 2009*). The TRAF-C domain is a binding scaffold mediating TRAF6 interaction with various effectors (*Ye et al., 2002*). Moreover, the CC domain has also been shown to be indispensable for TRAF6 oligomerization and subsequent activation of TAK1 complex (*Wang et al., 2006*). Recently, the CC domain has been shown to enhance TRAF6 activation through interacting with E2 Ubc13 (*Hu et al., 2017*). However, little is known about how the function of TRAF6 CC domain is regulated under inflammatory conditions. Herein, we provide compelling evidence that PIMT-mediated methylation of TRAF6 in the CC-domain represents a novel mechanism in the regulation of TRAF6/NF-κB signaling in vascular ECs. Overexpression of PIMT markedly impeded TRAF6 oligomerization and autoubiquitination, which is known to be a key process in the initiation of NF-κB activation in ECs. Furthermore, we identified Asn residue 350 in the CC domain of TRAF6

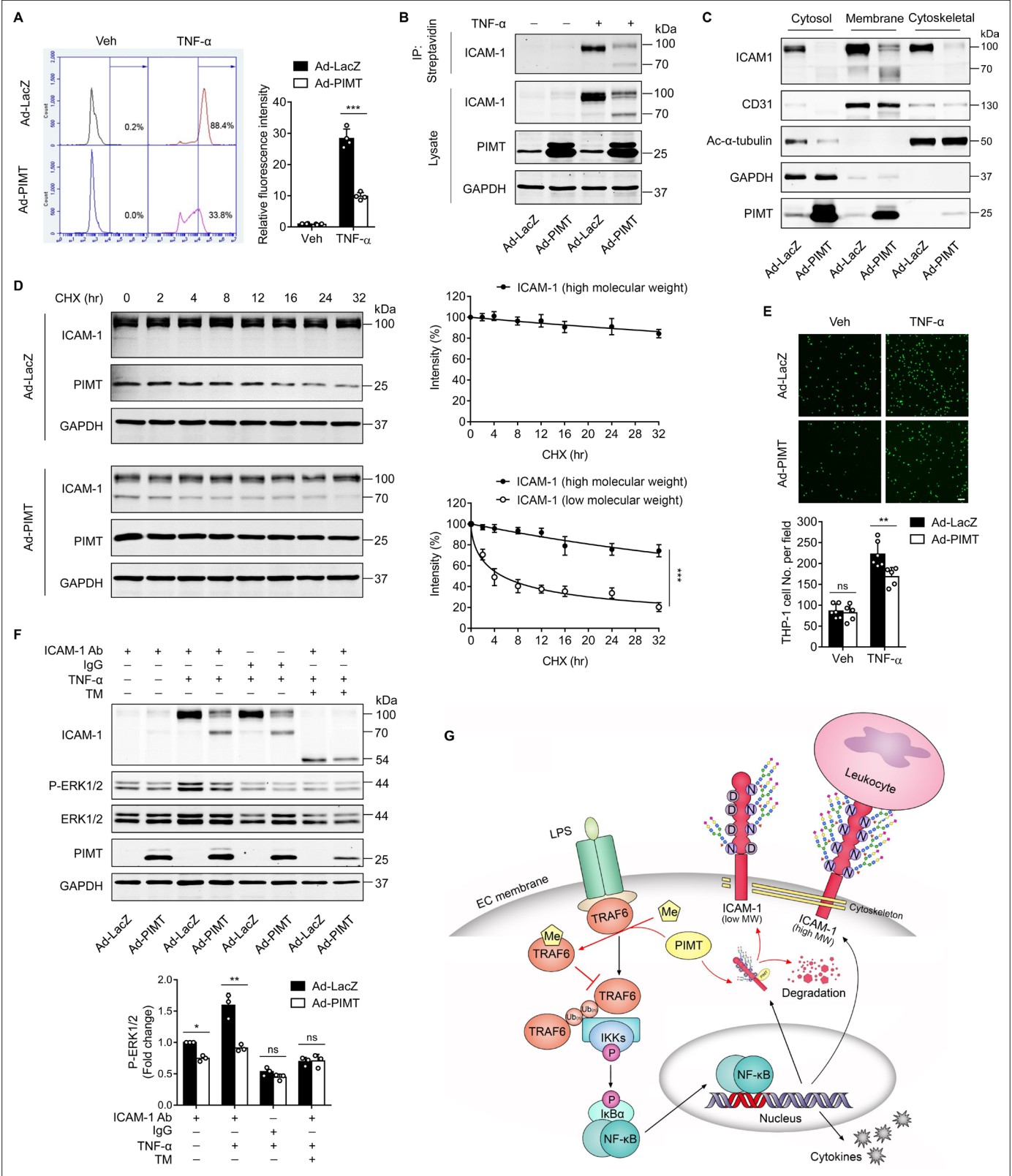

**Figure 6.** PIMT inhibits ICAM-1-mediated signaling events in ECs. (**A**) Human umbilical vein endothelial cells (HUVECs) transduced with indicated adenovirus (moi = 10, 48 hr) were stimulated with either vehicle or TNF-α (10 ng/ml) for 12 hr. Left, expression of ICAM-1 on the surface of HUVECs was labeled by fluorescent antibody and determined by flow cytometry. Right, the statistical summary of fluorescence intensity. ***p<0.001, two-way ANOVA coupled with Tukey's post hoc test, and n=4. (**B**) HUVECs were transduced with Ad-LacZ or Ad-PIMT (moi = 10) for 48 hr and then stimulated with

*Figure 6 continued on next page*

*Figure 6 continued*

TNF-α (10 ng/ml) for 12 hr and labeled with biotin. Precipitation was performed using streptavidin agarose beads, followed by western blot to detect the cell surface ICAM-1. (**C**) Western blot of ICAM-1 in different cellular fractions. HUVECs were fractionated after indicated virus transduction (moi, 10) and TNF-α (10 ng/ml, 12 hr) incubation. GAPDH, CD31, and acetyl-α-tubulin were used as cytosol, membrane, and cytoskeleton markers, respectively. (**D**) HUVECs were transduced with Ad-LacZ or Ad-PIMT (moi = 5, 48 hr) and stimulated with TNF-α (10 ng/ml) for 12 hr to induce both ICAM-1 bands, then subjected to the treatment with 10 μM cycloheximide (CHX) for indicated intervals. The half-life of ICAM-1 was determined by western blot (n=3), \*\*\*p<0.001, unpaired two-sided t test. (**E**) HUVECs transduced with adenovirus expressing LacZ or PIMT were stimulated with TNF-α (10 ng/ml, 12 hr) and then incubated with calcein-AM-labeled THP-1 for additional 1 hr. After washing, attached THP-1 cells were visualized and counted on an inverted fluorescent microscopy (n=6), \*\*p<0.01, and two-way ANOVA coupled with Tukey's post hoc test. Bars, 50 μm. (**F**) HUVECs pre-incubated with or without tunicamycin (TM, 2.5 μM) for 1 hr were exposed to indicated treatments and then subjected for ICAM-1 ligation using anti-ICAM-1 antibody followed by secondary antibody incubation. IgG were used as a negative cross-linking control. Phosphorylated and total $ERK_{1/2}$ were determined by western blot and quantified by densitometric analysis (n=3), \*p<0.05, \*\*p<0.01, and two-way ANOVA coupled with Tukey's post hoc test. (**G**) Schematic diagram of suppressing LPS-stimulated endothelial activation by PIMT.

The online version of this article includes the following source data for figure 6:

**Source data 1.** Effect of PIMT on the membrane expression of ICAM-1.

**Source data 2.** Effect of PIMT on the levels of ICAM-1 in different cellular fractions.

**Source data 3.** Effect of PIMT on ICAM-1 protein stability.

**Source data 4.** Effect of PIMT on $ERK_{1/2}$ activation after ICAM-1 ligation.

---

as the site of TRAF6 methylation by PIMT in ECs. Whether PIMT affects TRAF6 function through acting on Asn/Asp residues in other domains remains unknown and needs to be further investigated.

Deamidation of Asn is the first step in the formation of PIMT recognizable intermediates, such as isoAsp, succinimide, and methyl-isoAsp, and represents an important mechanism by which cells recognize deranged proteins during development and/or aging of cells (*Mishra and Mahawar, 2019*; *Reissner and Aswad, 2003*; *Lee et al., 2012*). Our data demonstrated that Asn substitution with Ala impaired TRAF6 susceptibility to PIMT-mediated inhibition but did not alter intrinsic functions, strongly suggesting that PIMT catalyzed methylation, rather than deamidated end product repair, is the primary mechanism coordinating TRAF6 homeostasis. Interestingly, we found that the N350 residue in the CC domain is localized in a loop site adjacent to TRAF-C domain, which is exposed to the protein surface and susceptible to deamidation (*Figure 4—figure supplement 1C*; *Ye et al., 2002*). Based on these findings, we hypothesize that PIMT-mediated N350 methylation serves as an intrinsic negative regulator of TRAF6 biological activity. It is worth noting that our study primarily focused on functional interactions of PIMT1 and TRAF6 because TRAF6 is critically implicated in LPS-induced endothelial activation and acute lung injury (*Chen, 2005*; *Martin and Wurfel, 2008*). Nevertheless, our data also demonstrated a weak interaction between PIMT and TRAF2. Since a conservative deamidation susceptible Asn site is identified in the TRAF-C domain of TRAF2, we cannot rule out the possibility that PIMT1-mediated O-methylation is involved in inhibiting TRAF2 activation. Furthermore, whether PIMT affects TRAF6 function through modification of Asn/Asp residues in other domains remains unknown and warrants further investigation.

ECs are regarded as sentinels for sensing invading pathogens and triggering innate immune system activation; however, hyperactive inflammatory responses by ECs can have detrimental effects on the host. Hence, confining inflammatory reactions is imperative for governing the magnitude and duration of endothelial activation and immune responses. Indeed, several molecules, such as IκBs, deubiquitinase A20, transcriptional co-factor CITED2, and protein chaperone cyclophilin J, have been identified as critical negative regulators of inflammatory signals under various conditions (*Boone et al., 2004*; *Lou et al., 2011*; *O'Dea and Hoffmann, 2010*; *Sheng et al., 2018*). LPS treatment should cause TRAF6 to become more prone to spontaneous modifications. This is because LPS is known to trigger production of reactive oxygen species (ROS; *Hsu and Wen, 2002*), and ROS is known to induce isoAsp generation (*D'Angelo et al., 2001*). In addition to altering the backbone, isoAsp modifications will have effects on protein conformation and PTMs, thus having either positive (e.g. p53) or negative (e.g. tubulin) downstream effects on protein functions (*Lee et al., 2012*; *Biterge et al., 2014*). In our study, we found that PIMT knockdown enhances TRAF6/NF-κB activation, suggesting that isoAsp modifications serve to overactivate TRAF6 function. IsoAsp repair is traditionally recognized as a spontaneous process. However, recent studies show that O-methylation of isoAsp occurs in nature, including on proteins such as p53 and histone H4 (*Lee et al., 2012*; *Biterge et al., 2014*), and is important for exerting biological functions. Notably, a recent publication demonstrated that O-methylation of p53 is not a transient event and is maintained on this protein for at least 24 hr (*Lee et al., 2012*).

LPS treatment did not alter PIMT levels under either in vitro or in vivo conditions but significantly augmented its interaction with TRAF6 in different EC types, suggesting that the dynamic formation of PIMT/TRAF6 protein complexes constitutes an important negative feedback mechanism in resolving LPS inflammation. It should be noted that in addition to ECs, PIMT is also expressed in other type of cells, such as macrophage and epithelial cells in the lung. Future studies using tissue specific PIMT knockout mice will help to determine the relative contribution of PIMT in different cell types to ALI.

As a hallmark of endothelial activation, cell-surface expression of adhesion molecules mediates the interaction of circulating leukocytes with endothelium in response to proinflammatory signals (*Hunt and Jurd, 1998*). Adhesion molecules, such as ICAM-1 and VCAM-1, are heavily N-glycosylated, and N-glycosylation defects have been shown to impair expression and membrane localization of adhesion molecules and subsequent intracellular, cell–cell, and cell–matrix recognition events in inflammatory responses (*Chen et al., 2014b*; *Scott and Patel, 2013*). N-glycan attaches only to Asn and presents in an N-x-S/T consensus sequence. At this time, little is known about how N-glycosylation of adhesion molecules is regulated under inflammatory conditions in ECs. Here, we show that PIMT affects inflammatory responses through modulating its N-glycosylation of ICAM-1. We found that PIMT differentially regulates the N-glycosylation of ICAM-1 without altering the global endothelial glycocalyx. To further investigate molecular mechanisms underlying inhibition of adhesion molecule N-glycosylation by PIMT, we chose to determine the structure of the N-glycans of ICAM-1 using mass spectrometry. ICAM-1 possesses eight N-glycosylation sites. We found that PIMT inhibits ICAM-1 N-glycosylation at four specific Asn sites (N145, N183, N267, and N385). Furthermore, our mass spectrometric data show that Asn residues at these sites are switched to Asp residues by PIMT, thus preventing N-glycosylation. Our results also suggest that the enzymatic activity of PIMT may be necessary for the O-methylation of isoAsp derived from Asn to form Asp in ICAM-1. In addition to the decreasing ICAM-1 protein stability, as shown in this study, we suspect that deficiency of ICAM-1 N-glycosylation may directly impact the function of ICAM-1, despite our data showing that LMW form of ICAM-1 can still localize to the cell membrane. As shown in *Figure 5H*, the N145 and N183 sites are localized in the hydrophilic BC and EF loops of ICAM-1 domain 2 (D2) upon structural dissection (*Casasnovas et al., 1998*). Based on the fact that ICAM-1 D2 sustains the structural and LFA1 binding integrity of D1 (*Stanley et al., 2000*), we speculate that PIMT-induced 'site mutation' at these sites may impair the binding capacity of ICAM-1 to its ligand, and this hypothesis was partially testified in our ICAM-1 antibody cross-linking experiments. Furthermore, lack of ICAM-1 glycosylation at N267 in the D3 domain may also affect interactions with macrophage-1 antigen (Mac-1; *Diamond et al., 1993*). In this regard, our study provides novel insights into how PIMT-mediated N-glycosylation regulates the functional interaction of ECs and immune cells in vascular inflammation. The reason why PIMT specifically acts on these four specific Asn sites remains elusive. It is attempted to speculate that isoAsp repair and maintenance, just like other protein modifications, may occur in a context and cell type dependent fashion. In our IP experiment, only small fractions of PIMT co-IP'd with LMW ICAM-1. Although we suspect this could be explained by a 'hit and run' mechanism. Furthermore, we cannot exclude the possibility that PIMT may affect any one of the many steps along the pathway to protein glycosylation, such as ERQC or glucoside attachment. Elucidation of molecular mechanisms underlying PIMT-induced glycosylation of VCAM-1 in ECs is ongoing.

In summary, our work has identified novel mechanisms in the control of agonist-induced EC activation by PIMT. We demonstrate that PIMT-catalyzed methylation directly restricts TRAF6 activation to maintain TRAF6 in the default static state. Moreover, PIMT provides a 'double check' mechanism to avoid endothelial activation by limiting ICAM-1 membrane expression and intracellular signaling through transcriptional and post-translational controls (*Figure 6G*). Altogether, these results suggest that targeted activation of PIMT may provide a novel therapeutic strategy for treating a wide range of inflammatory vascular disorders.

# Materials and methods

## Key resources table

| Reagent type (species) or resource | Designation | Source or reference | Identifiers | Additional information |
|---|---|---|---|---|
| Genetic reagent (*M. musculus*) | B6;129S4-*Pcmt*1[tm1Sc]l/J | The Jackson Laboratory | #023343 | |

*Continued on next page*

*Continued*

| Reagent type (species) or resource | Designation | Source or reference | Identifiers | Additional information |
|---|---|---|---|---|
| Genetic reagent (*M. musculus*) | C57BL/6J | The Jackson Laboratory | #000664 | |
| Cell line (*Homo-sapiens*) | HUVEC | Gibco | C0155C | |
| Cell line (*Homo-sapiens*) | HLMVEC | Lonza | CC-2527 | |
| Cell line (*Homo-sapiens*) | HEK-293T | ATCC | CRL-3216 | |
| Cell line (*Homo-sapiens*) | EA.hy926 | ATCC | CRL-2922 | |
| Cell line (*Homo-sapiens*) | THP-1 | ATCC | TIB-202 | |
| Cell line (*Homo-sapiens*) | HEK-293 | ATCC | CRL-1573 | |
| Antibody | Rabbit polyclonal anti-PCMT1 | ABclonal | A6684 | IF (1:200), IP (1:100) WB (1:1000) |
| Antibody | Rabbit polyclonal anti-TRAF6 | ABclonal | A0973 | IP (1:100) WB (1:1000) |
| Antibody | Rabbit polyclonal anti-ICAM-1 | ABclonal | A5597 | IP (1:100) FCM (1:200) WB (1:1000) |
| Antibody | Rabbit polyclonal anti- phospho-TAK1-T187 | ABclonal | AP1222 | WB (1:1000) |
| Antibody | Rabbit polyclonal anti-HA | ABclonal | AE036 | WB (1:1000) |
| Antibody | Rat monoclonal anti-PECAM1 | BD Pharmingen | 550274 | IF (1:200) |
| Antibody | Mouse monoclonal anti-Flag | Sigma-Aldrich | F3165 | WB (1:2000) |
| Antibody | Mouse monoclonal anti-HA | Sigma-Aldrich | H9658 | WB (1:2000) |
| Antibody | Mouse monoclonal anti-c-Myc | Sigma-Aldrich | C3956 | WB (1:2000) |
| Antibody | Rabbit polyclonal anti-GAPDH | Proteintech | 10494–1-AP | WB (1:1000) |
| Antibody | Rabbit polyclonal anti- Tubulin | Proteintech | 11224–1-AP | WB (1:1000) |
| Antibody | Rabbit polyclonal anti- TRAF6 | Proteintech | 66498–1-Ig | WB (1:1000) |
| Antibody | Rabbit monoclonal anti- p65 | CST | #8242 | IF (1:200) WB (1:1000) |
| Antibody | Rabbit monoclonal anti- phospho-IKKα/β | CST | #2697 | WB (1:1000) |
| Antibody | Rabbit monoclonal anti- IKKβ | CST | #8943 | WB (1:1000) |
| Antibody | Rabbit monoclonal anti- phospho-IκBα | CST | #2859 | WB (1:1000) |
| Antibody | Rabbit polyclonal anti- IκBα | CST | #9242 | WB (1:1000) |
| Antibody | Rabbit monoclonal anti- phospho-Erk1/2 | CST | #4370 | WB (1:1000) |
| Antibody | Rabbit monoclonal anti- Erk1/2 | CST | #4695 | WB (1:1000) |
| Antibody | Rabbit monoclonal anti- acetyl-α-Tubulin | CST | #5335 | WB (1:1000) |
| Antibody | Rabbit polyclonal anti- Lamin A/C | Santa Cruz | sc-20681 | WB (1:1000) |
| Antibody | Rabbit polyclonal anti- ICAM-1 | Santa Cruz | sc-7891 | WB (1:1000) |
| Antibody | Mouse monoclonal anti- VCAM1 | Santa Cruz | sc-13160 | WB (1:1000) |
| Antibody | Mouse monoclonal anti-PCMT1 | Santa Cruz | sc-100977 | WB (1:1000) |

*Continued*

| Reagent type (species) or resource | Designation | Source or reference | Identifiers | Additional information |
|---|---|---|---|---|
| Antibody | Mouse monoclonal anti-GAPDH | Santa Cruz | sc-32233 | WB (1:2000) |
| Antibody | Donkey polyclonal anti-Rabbit | LI-COR | 926–68073 | IRDye 680RD, WB (1:10000) |
| Antibody | Goat polyclonal anti-Mouse | LI-COR | 926–32210 | IRDye 800CW, WB (1:10000) |
| Antibody | Donkey polyclonal anti-mouse | Invitrogen | A21202 | Alexa Flour 488, IF (1:1000) |
| Antibody | Donkey polyclonal anti-rat | Invitrogen | A21208 | Alexa Flour 488, IF (1:1000) |
| Antibody | Goat polyclonal anti-rabbit | Invitrogen | A21428 | Alexa Flour 555, IF (1:1000) |
| Antibody | Goat polyclonal anti-rabbit | Invitrogen | A32733 | Alexa Flour 647, FCM (1:1000) |
| Recombinant DNA reagent | pFLAG-CMV-2-PIMT | PMID:23647599 | | |
| Recombinant DNA reagent | pcDNA3-FLAG-TRAF6 | Addgene | #66929 | |
| Recombinant DNA reagent | pcDNA3-FLAG-TRAF2 | Addgene | #66931 | |
| Recombinant DNA reagent | pcDNA-FLAG-IKKβ | Addgene | #23298 | |
| Recombinant DNA reagent | FLAG-IKKβ-S177E/S181E | Addgene | #11105 | |
| Recombinant DNA reagent | HA-Ubiquitin | Addgene | #18712 | |
| Recombinant DNA reagent | pLKO-PIMT | Sigma-Aldrich | TRCN0000036401, TRCN0000036403 | |
| Recombinant DNA reagent | pLKO-TRAF6 | Sigma-Aldrich | TRCN0000007348 | |
| Recombinant DNA reagent | pLKO-non target control | Sigma-Aldrich | SHC016 | |
| Peptide, recombinant protein | Human TNF-α | PeproTech | 300–01 A | |
| Peptide, recombinant protein | Human IL-1β | PeproTech | 200-01B | |
| Commercial assay or kit | Mouse TNF-α ELISA | R&D | DY-410 | |
| Commercial assay or kit | Mouse IL-6 ELISA | R&D | DY-406 | |
| Commercial assay or kit | Mouse Ccl2 ELISA | R&D | DY-479 | |
| Chemical compound and drug | CellTracker | Invitrogen | C2925 | |
| Chemical compound and drug | DAPI | Invitrogen | D1306 | |
| Chemical compound and drug | LPS | Santa Cruz | sc-3535 | |
| Chemical compound and drug | Cycloheximide | Sigma-Aldrich | 01810 | |
| Chemical compound and drug | Anti-FLAG M2 Magnetic Beads | Sigma-Aldrich | M8823 | |
| Chemical compound and drug | Protein standards | Bio-Rad | #1610374 | |
| Chemical compound and drug | Protein standards | Genscript | M00624 | |

## Mice

PIMT heterozygous knockout mice (B6;129S4-*Pcmt1*tm1Scl/J, #023343) were purchased from The Jackson Laboratory and bred with WT counterparts (C57BL/6J, #000664, JAX). Genotyping PCR was performed using genomic DNA by tail snipping. The WT (5'-AGTGGCAGCGACGGCAGTAACAGCG G-3' and 5'-ACCCTCTTCCCATCCACATCGCCGAG-3') and knockout primer sets (5'-CGCATCGAG CGAGCACGTACTCGG-3' and 5'-GCACGAGGAAGCGGTCAGCCCATTC-3') targeting *Pcmt1* exon 1 or neomycin cassette as described elsewhere before (*Dai et al., 2013*). Mixed male and female mice aged 8–10 weeks were used in all studies. Animal protocols were approved by the Institutional Animal Care and Use Committee at Thomas Jefferson University before initiation of any studies. Euthanasia

was performed in accordance with recommendations of the American Veterinary Medical Association Panel on Euthanasia. A block randomization method was used to assign experimental animals to groups on a rolling basis to achieve adequate sample number for each experimental condition. Experiments were performed and analyzed with operators blinded to the genotype and treatment allocation. No exclusion criteria were pre-determined, and no animals were excluded. Sample sizes were calculated with a priori analysis by G-power 3.1.9.2 software. For a twofold change in a parameter, with an SD up to 50% of the mean, there is an 85% probability that six mice per group will be needed to detect a difference at 5% level of significance in an unpaired two-tailed Student t test or ANOVA. Thus, more than six mice were used in most of the animal studies. Representative images were chosen to most accurately represent the group mean/average across all the available data.

## Cell culture

HUVECs (Gibco, C0155C) and HLMVECs (Lonza, CC-2527) were used in our studies. Identity of endothelial cells was confirmed by immunostaining of CD31, von Williebrand Factor VIII, and positive for acetyated low density lipoprotein uptake. Cells were cultured in complete endothelial cell medium (ScienCell, 1001) and discarded after three passages. HEK-293T (ATCC, CRL-3216), EA.hy926 (ATCC, CRL-2922), and HEK-293 (ATCC, CRL-1573) were maintained in DMEM medium (Corning, 10013CV), and THP-1 cells (ATCC, TIB-202) were cultures in RPMI-1640 medium (Corning, 10–040 CM). All medium was supplemented with fetal bovine serum (FBS; Gibco, 1082147) and penicillin/streptomycin (Corning, 30–002 CI). Cell identities were confirmed at the commercial source (ATCC) using satellite tandem repeat profiling and were tested to be free from mycoplasma.

## Reagents and antibodies

Anti-PCMT1 (A6684), anti-TRAF6 (A0973), anti-ICAM-1 (A5597), anti-phospho-TAK1-T187 (AP1222), and anti-HA (AE036) antibodies were purchased from ABclonal Technology. Anti-PECAM1 (550274) antibody was purchased from BD Pharmingen. Anti-Flag (F3165), anti-HA (H9658), anti-c-Myc (C3956) antibodies, TM (11089-65-9), and Cycloheximide (C7698) were obtained from Sigma-Aldrich. Anti-GAPDH (10494–1-AP), anti-alpha Tubulin (11224–1-AP), and anti-TRAF6 (66498–1-Ig) antibodies were acquired from Proteintech Group. Anti-p65 (#8242), anti-phospho-IKKα/β (#2697), anti-IKKβ (#8943), anti-phospho-IκBα (#2859), anti-IκBα (#9242), anti-phospho-p44/42 MAPK (Erk1/2; #4370), anti-p44/42 MAPK (Erk1/2; #4695), and anti-acetyl-α-Tubulin (Lys40; #5335) antibodies were purchased from Cell Signaling Technology. Anti-Lamin A/C (sc-20681), anti-ICAM-1 (sc-7891), anti-VCAM1 (sc-13160), anti-PCMT1 (sc-100977), anti-GAPDH (sc-32233) antibodies, and LPS (sc-3535) were acquired from Santa Cruz Biotechnology. IRDye 680RD Donkey anti-Rabbit (926–68073), 800CW Goat anti-Mouse (926–32210) antibodies, and 680RD Streptavidin (926–68079) were from obtained from LI-COR Bioscience. Alexa Flour 488 donkey anti-mouse (A21202), 488 donkey anti-rat (A21208), 555 goat anti-rabbit (A21428), 647 goat anti-rabbit (A32733) antibodies, CellTracker Green CMFDA Dye (C2925), and DAPI (4′,6-diamidino-2-phenylindole; D1306) were from purchased from Invitrogen. EZ-Link Sulfo-NHS-LC-LC-Biotin (#21135) and Streptavidin Agarose (#20357) were obtained from Pierce. Biotinylated PHA-L (B-1115) was purchased from Vector Laboratories. Recombinant Human TNF-α (300–01 A) and IL-1β (200-01B) were acquired from PeproTech.

## Plasmids

pFLAG-CMV-2-*PIMT* construct was described previously (*Yan et al., 2013*). Myc-PIMT was generated by cloning the PIMT fragment into pCS2–6×Myc plasmid. pcDNA3-FLAG-*TRAF6* (#66929), pcDNA3-FLAG-*TRAF2* (#66931), pcDNA-FLAG-*IKKβ* (#23298), FLAG-*IKKβ-S177E/S181E* (#11105), and HA-Ubiquitin (#18712) were acquired from Addgene. HA-TRAF6 was generated by cloning the *TRAF6* fragment into pcDNA3-HA plasmid, and HA or Myc tagged TRAF6 truncations were made from pcDNA3-FLAG-*TRAF6* or pcDNA3-HA-*TRAF6* for direct transfection. Flag-TAK1 was generated by cloning the *TAK1* fragment into pFLAG-CMV-2 plasmid. PIMT and TRAF6 mutants were made using the QuikChange II Site-Directed Mutagenesis Kit (Agilent, #200523). Primers for generating mutants were: *PIMT* motif I deletion mutant, 5′-CAAAACATGCAGTAAGGATTTTAGCTCCTTCATGCAACTGATC-3′ (F) and 5′-GATCAGTTGCATGAAGGAGCTAAAATCCTTACTGCATGTTTTG-3′ (R); *TRAF6* D339A, 5′-CGATTTCAGCAACTTTGGCCTCAAGGGGTTCGAA-3′ (F) and 5′-CATTCGAACCCCTTGAGGCCAAAGTTGCTGAAATCG-3′ (R); *TRAF6* N350A, 5′-CAATCTTCCAAATATAAATTCCAGCGCACTGCTGTGC

TTCGATTTCAG-3′ (F) and 5′-CTGAAATCGAAGCACAGCAGTGCGCTGGAATTTATATTTGGAAGA TTG-3′ (R). All original constructs in this study were verified by DNA sequencing.

## Bronchoalveolar lavage

Bronchoalveolar lavage was performed by cannulating the trachea with a blunt 22-gauge needle and performing whole lung lavage with 1 ml PBS as described before (*Zhu et al., 2019*). Total cell counts were determined using a TC20 automated cell counter (Bio-Rad). After centrifugation, BALF supernatant was subjected to protein concentration measurement by BCA assay (Pierce, #23225). BALF neutrophil counts were performed after cyto-centrifugation of cells onto glass slides and staining with Kwik-Diff solutions (Thermo Scientific, #9990701).

## Lung histology

Lung histology was performed on paraformaldehyde fixed tissues embedded in paraffin wax. Sections (6 µm) were placed on positively charged glass slides and then deparaffinized for hematoxylin and eosin staining. Tissues were visualized with a light microscope (Nikon).

## Enzyme-linked immunosorbent assay

Enzyme-linked immunosorbent assay for TNF-α (DY-410), IL-6 (DY-406), and Ccl2 (DY-479) was performed according to the manufacturer's manual (R&D Systems). Briefly, samples and standards were added in various dilutions to the plate coated with capture antibody and then labeled with biotinylated detection antibody. After incubation with streptavidin-peroxidase and substrate, protein concentration was determined by absorbance at 450 nm in a plate reader (Biotek Instrument).

## Adenovirus construction and purification

Adenoviruses were generated using RAPAd adenoviral expression systems (Cell Biolabs) according to the manual. Briefly, pacAd5 CMVK-NpA shuttle vector containing gene of interest and pacAd5 9.2–100 backbone vector were linearized using PacI digestion and then co-transfected to HEK-293 cells. Crude viral lysates were harvested from the adenovirus-containing cells by three freeze/thaw cycles. The viral supernatants were propagated in HEK-293 cells to amplify and purified through CsCl density-gradient ultracentrifugation, followed by dialysis into 20 mM Tris buffer and supplement with 10% glycerol for stocking. Viral titers were determined using the Adeno-X Rapid Titer Kit (Takara Bio, 632250).

## Transient transfection

DNA was transfected using polyethylenimine (PEI). In general, cells were seeded to 80% confluence at transfection. Plasmid DNA was mixed with PEI (1 mg/ml) at 1:3 in Opti-MEM medium (Gibco, 11058021) and added to cells. Medium was changed after 6 hr, and cells were harvested after 48 hr incubation.

## Lentivirus infection

Stable cell lines were made by lentivirus transduction, as previously described (*Chen et al., 2014a*) with slight modifications. pLKO-*PIMT* (TRCN0000036401 and TRCN0000036403), *TRAF6* (TRCN0000007348) constructs, and non-target shRNA control (SHC016) were from Sigma-Aldrich. Lentiviral pLKO constructs were transfected with packaging and envelope plasmids to HEK293T cells. Viral supernatant was harvested 48 hr and 72 hr post-transfection, filtered through a 0.45 µm filter and then added to cultured cells in medium supplemented with 10 µg/ml polybrene (Sigma-Aldrich). 24 hr after infection, the virus-containing medium was removed and replaced with fresh medium. The infected cells were selected with puromycin (1 µg/ml) for ~1 week.

## Luciferase reporter assay

NF-κB firefly luciferase reporter vectors (pNF-κB-TA-Luc) and Renilla luciferase control reporter vectors (pRL-TK), together with indicated expression plasmids, were transfected into cells pre-seeded in 24-well plates. Luciferase level was measured as described previously by a luminescence microplate reader (BioTek). In brief, cells were lysed at 48 hr, and firefly luciferase was detected using 2×GoldBio Luciferase Assay Buffer containing 200 mM Tris (pH 7.8), 10 mM MgCl$_2$, 0.5 mM Coenzyme A (CoA),

0.3 mM ATP, and 0.3 mg/ml Luciferin (Gold Biotechnology, #LUCK). Renilla luciferase was assessed using 2 µM coelenterazine in DPBS(phosphate-buffered saline). For LPS stimulated reporter activity, cells were incubated for 10 hr with designated treatments and lysed.

## Western blot

Cells were lysed in RIPA buffer containing 50 mM Tris (pH 8.0), 0.5 mM EDTA, 150 mM NaCl, 1% NP-40, and 1% sodium dodecyl sulfate (SDS), supplemented with protease and phosphatase inhibitors. The lysates were agitated, and the supernatants were denatured and subjected to SDS-PAGE. After transfer, nitrocellulose membranes were blocked with 5% BSA at room temperature for 1 hr. Primary antibodies were incubated at 4°C overnight. After washing, membranes were incubated with IRDye secondary antibody conjugates at room temperature for 1 hr. Membrane were then washed and imaged by the Odyssey infrared imaging system (LI-COR, 9120).

## Immunofluorescence

Cells plated on eight-well Chamber Slides (Thermo Scientific, 154534) were fixed with 4% formaldehyde and permeabilized with 0.2% Triton-X 100 at room temperature. After blocking with 5% BSA for 1 hr, slides were incubated with primary antibodies overnight at 4°C, followed by exposure to Alexa Fluor-conjugated secondary antibodies for 1 hr at room temperature. After staining with DAPI (1 µg/ml), wells were removed, and slides were mounted with 50% glycerol and imaged with a fluorescent confocal microscope (Nikon). For staining of lung frozen sections, lungs were inflated with OCT(optimal cutting temperature) before snap freezing and sectioning at a thickness of 5 µm, followed by immunostaining.

## Immunoprecipitation

Cells were lysed in IP buffer containing 20 mM Tris (pH 8.0), 137 mM NaCl, 2 mM EDTA, 1% NP-40, and 10% glycerol, supplemented with protease and phosphatase inhibitors. For endogenous IP, supernatants were pre-cleaned with IgG and Dynabeads (Thermo Fisher Scientific) then incubated with antibodies at 4°C overnight with rotation, and indicated beads were added for additional 2 hr on the next day. For exogenous IP, the supernatants were incubated with Anti-FLAG M2 Magnetic Beads (Sigma-Aldrich) and rotated at 4°C overnight. The IP was washed three times with lysis buffer, collected by magnet, and then boiled with SDS reducing loading dye. Samples were analyzed by western blot.

## Subcellular fractionation

Nucleus and cytosol isolation was performed with the NE-PER Nuclear and Cytoplasmic Extraction Reagents (Thermo Scientific, #78835) according to manufacturer instructions. Nuclear extracts were used for EMSA. Extraction of cell membrane and cytoskeleton fractions was carried out using the Subcellular Protein Fractionation Kit for Cultured Cells (Thermo Scientific, #78840) in line with the manual.

## Electrophoretic mobility shift assay

EMSA was performed with IRDye 700 NFκB Consensus Oligonucleotide (LI-COR, 829–07924) according to the manufacturer instructions with slight modifications. Briefly, 10 µg nuclear extracts prepared from HUVECs were incubated with 40 fmol labeled NF-κB oligonucleotides at room temperature for 30 min, and DNA-protein complexes were separated in 4% native polyacrylamide gel containing 50 mM Tris (pH 7.5), 0.38 M glycine, and 2 mM EDTA, the shift was detected by the Odyssey infrared imaging system. The binding specificity was determined using 50-fold unlabeled DNA duplex or anti-p65 antibody.

## Flow cytometry

Virus transduced HUVECs were treated with TNFα for 12 hr to induce ICAM-1 and VCAM-1 expression. HUVECs were then trypsinized and resuspended in fluorescence-activated cytometry sorting (FACS) buffer containing 10% FBS and 1% sodium azide in PBS, followed by incubation with indicated primary antibodies for 1 hr on ice. After washing, cells were incubated with Alexa fluor conjugated fluorescent secondary antibodies for 30 min on ice in the dark. Then samples were washed again and

resuspended in FACS buffer. For each sample, $1.5 \times 10^4$ cells were analyzed by a BD Accuri C6 flow cytometer (BD Bioscience).

## Monocyte adhesion assay

HUVECs were seeded in 24-well plates pre-coated with 0.1% gelatin. After viral transduction and exposure to inflammatory cytokines, $2.5 \times 10^5$ THP-1 cells labeled with calcein CellTracker Green CMFDA Dye (Invitrogen) were added to each well in serum-free media and incubated for 30 min at 37°C. After washing, the number of attached THP-1 cells was counted on the EVOS FL Auto Imaging System (Invitrogen).

## Quantitative real-time RT-PCR

qRT-PCR was performed as described previously. Total RNA was extracted from cells with TRIzol reagent (Invitrogen), and cDNA was synthesized using the High-Capacity cDNA Reverse Transcription Kit (Applied Biosystems). RT-PCR was performed using the PowerUp SYBR Green Master Mix (Applied Biosystems) on a CFX Connect Real-Time PCR Detection System (Bio-Rad), and results were calculated by the comparative cycling threshold (Ct) quantification method. The gene encoding GAPDH was served as an internal control for the total amount of cDNA. The primer sequences used were described as follows: Human *ICAM-1*, 5′-CTTCGATCCCAAGGTTTCCAA-3′ (F) and 5′-TCGACACAA AGGATTTCGTAAGG-3′ (R); *VCAM-1*, 5′-GGGAAGATGGTCGTGATCCTT-3′ (F) and 5′-TCTGGGGGTG GTCTCGATTTTA-3′ (R); *TNFα*, 5′- CCTCTCTCTAATCAGCCCTCTG-3′ (F) and 5′- GAGGACCTGGGA GTAGATGAG-3′ (R); *IL-1β*, 5′-ATGATGGCTTATTACAGTGGCAA-3′ (F) and 5′-GTCGGAGATTCGT AGCTGGA-3′ (R); *IL-6*, 5′-ATGAGGAGACTTGCCTGGTGAA-3′ (F) and 5′-AACAATCTGAGGTGCCC ATGCTAC-3′ (R); *Ccl-2*, 5′-CAGCCAGATGCAATCAATGCC-3′ (F) and 5′-TGGAATCCTGAACCCAC TTCT-3′ (R); *GAPDH*, 5′-ACAACTTTGGTATCGTGGAAGG-3′ (F) and 5′-GCCATCACGCCACAGTTTC-3′ (R); mouse *ICAM-1*, 5′-GTGATGCTCAGGTATCCATCCA-3′ (F) and 5′-CACAGTTCTCAAAGCAC AGCG-3′ (R); *TNFα*, 5′- CCCTCACACTCAGATCATCTTCT-3′ (F) and 5′- GCTACGACGTGGGCTACAG-3′ (R); *IL-1β*, 5′- GCAACTGTTCCTGAACTCAACT-3′ (F) and 5′- ATCTTTTGGGGTCCGTCAACT-3′ (R); *IL-6*, 5′- TAGTCCTTCCTACCCCAATTTCC-3′ (F) and 5′- TTGGTCCTTAGCCACTCCTTC-3′ (R); *Ccl-2*, 5′- TTAAAAACCTGGATCGGAACCAA-3′ (F) and 5′- GCATTAGCTTCAGATTTACGGGT-3′ (R); *GAPDH*, 5′- AGGTCGGTGTGAACGGATTTG-3′ (F) and 5′- TGTAGACCATGTAGTTGAGGTCA-3′ (R).

## Ubiquitination assay

Detection of protein ubiquitination in cultured cells was conducted as described previously (*Choo and Zhang, 2009*). Basically, cells were lysed with 2% SDS, 150 mM NaCl, and 10 mM Tris (pH 8.0) supplemented with protease inhibitors. The lysates were boiled and sonicated and diluted with 150 mM NaCl, 10 mM Tris (pH 8.0), 2 mM EDTA, and 1% Triton. After incubation at 4°C with rotation, the supernatants were collected for anti-Flag IP.

## Protein sample preparation and nano LC-MS/MS

Gel band was destained with 100 mM ammonium bicarbonate/acetonitrile. The band was reduced in 10 mM dithiothreitol/100 mM ammonium bicarbonate and alkylated with 100 mM iodoacetamide in 100 mM ammonium bicarbonate. Proteins in the gel band were then digested with trypsin overnight and collected in supernatants. Additional gel peptides were extracted by 50% acetonitrile and 1% TFA(Trifluoroacetic acid). The supernatants were combined and dried, followed by reconstitution of 0.1% formic acid for mass spectrometry analysis.

Desalted peptides were analyzed on a Q-Exactive HF attached to an Ulimate 300 nano UPLC system (Thermo Scientific). Peptides were eluted with a 25 min gradient from 2 to 32% ACN(acetonitrile) and to 98% ACN over 5 min in 0.1% formic acid. Data dependent acquisition mode with a dynamic exclusion of 45 s was enabled. One full MS scan was collected with scan range of 350–1200 m/z, resolution of 70 K, maximum injection time of 50 ms, and AGC of 1e6. Then, a series of MS2 scans were acquired for the most abundant ions from the MS1 scan (top 15). An isolation window of 1.4 m/z was used with quadruple isolation mode. Ions were fragmented using higher-energy collisional dissociation with a collision energy of 28%.

Proteome Discoverer 2.3 (Thermo Scientific) was used to process the raw spectra. Database search criteria were as follows: (taxonomy *Homo sapiens*) carboxyamidomethylated (+57 Da) at cysteine

residues for fixed modifications; oxidized at methionine (+16 Da) residues; phosphorylation (+79.9 Da) at serine, threonine, and tyrosine residues; deamidation (+0.98 Da) and methylation (+14 Da) at Asn/Asp for variable modifications, two maximum allowed missed cleavage, 10 ppm MS tolerance, and a 0.02 Da MS/MS tolerance. Only peptides resulting from trypsin digestion were considered. The target-decoy approach was used to filter the search results, in which the false discovery rate was less than 1% at the peptide and protein level.

## Statistical analysis

Data were presented as mean ± SD and plotted using GraphPad Prism version 7.00 software. Two-tailed Student's t test was applied for comparison between two groups. Two-way ANOVA coupled with Tukey's post hoc test was applied for two independent variables. Significance was considered when the p value was less than 0.05.

## Acknowledgements

This work was funded by NIH grants (R01HL159168 and R01HL152703) and American Heart Association Established Investigator Award 16EIA27710023 (to JS).

## Additional information

### Funding

| Funder | Grant reference number | Author |
|---|---|---|
| American Heart Association | 16EIA27710023 | Jianxin Sun |
| National Heart, Lung, and Blood Institute | R01HL159168 | Jianxin Sun |
| National Heart, Lung, and Blood Institute | R01HL152703 | Jianxin Sun |

The funders had no role in study design, data collection and interpretation, or the decision to submit the work for publication.

### Author contributions

Chen Zhang, Conceptualization, Data curation, Formal analysis, Validation, Investigation, Visualization, Methodology, Writing - original draft; Zhi-Fu Guo, Data curation, Formal analysis, Investigation, Methodology; Wennan Liu, Data curation, Visualization, Methodology; Kyosuke Kazama, Louis Hu, Lu Wang, Visualization, Methodology; Xiaobo Sun, Validation, Methodology; Hyoungjoo Lee, Data curation, Methodology; Lin Lu, Conceptualization, Visualization, Methodology, We add Dr. Lin Lu as a coauthor because of his contribution to the experimental design and data interpretation of mass spectrometric analysis of ICAM-1 glycosylation; Xiao-Feng Yang, Ross Summer, Conceptualization, Visualization, Writing – review and editing; Jianxin Sun, Conceptualization, Resources, Data curation, Formal analysis, Supervision, Funding acquisition, Validation, Investigation, Visualization, Methodology, Writing - original draft, Project administration, Writing – review and editing

### Author ORCIDs

Chen Zhang (iD) http://orcid.org/0000-0002-8198-3090
Kyosuke Kazama (iD) http://orcid.org/0000-0001-6794-6982
Lu Wang (iD) http://orcid.org/0000-0002-8896-0115
Jianxin Sun (iD) http://orcid.org/0000-0001-5319-912X

### Ethics

This study was performed in strict accordance with the recommendations in the Guide for the Care and Use of Laboratory Animals of the National Institutes of Health. The protocol was approved by the institutional animal care and use committee (IACUC) of Thomas Jefferson University (protocol number: 01600).

## Decision letter and Author response

Decision letter https://doi.org/10.7554/eLife.85754.sa1
Author response https://doi.org/10.7554/eLife.85754.sa2

# Additional files

## Supplementary files

• MDAR checklist

## Data availability

All data generated or analysed during this study are included in the manuscript and supporting file; Source Data files have been provided for Figures 1-6.

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
