## [Editor Report]

A multidisciplinary approach combining biochemical, cellular, and Mass-spec methods shows that PIMT regulates immune responses. PIMT exerts its anti-inflammatory function by inhibiting TRAF6 and ICAM-1. This unexpected function of PIMT in suppressing endothelial activation could open new doors in developing therapeutic strategies against vascular diseases.

---

## [Decision Letter]

**Decision letter after peer review:**

[Editors’ note: the authors submitted for reconsideration following the decision after peer review. What follows is the decision letter after the first round of review.]

Thank you for submitting the paper "PIMT: a novel and potent suppressor of endothelial activation" for consideration by *eLife*. Your article has been reviewed by 3 peer reviewers, one of whom is a member of our Board of Reviewing Editors, and the evaluation has been overseen by a Senior Editor. The reviewers have opted to remain anonymous.

Comments to the Authors:

We are sorry to say that, after consultation with the reviewers, we have decided that this work will not be considered further for publication by *eLife* in its current form.

The consensus from the review was that although the identification of the role of PIMT in regulating vascular homeostasis is impactful, there were several major issues to be addressed to support the publication. Particularly, the reviewers felt that the premise for the role of isoAsp in pro-inflammatory function needs to be better established, the mass-spec data need to be presented more clearly, and the in vivo relevance of in vitro (over expression) results should be more clearly established (e.g., the effect on ICAM-I).

We, however, remain enthusiastic about the conceptual advancement of the work. Thus, we will be happy to consider an improved manuscript that has addressed these concerns.

*Reviewer #1 (Recommendations for the authors):*

The authors found that impairing PIMT (KD) exacerbated inflammation and vascular leakage upon LPS challenge. They report here that this is due to the lack of repair on damaged Asn on TRAF6, which resulted in inhibition of its oligomerization and ubiquitination (consequently preventing NF-κB transactivation). The authors also found that PIMT was linked to regulating ICAM-1 expression by inhibiting N-glycosylation. This study reveals a previously unappreciated function of PIMT in regulating innate immune responses- limiting inflammatory responses thus maintaining vascular homeostasis.

I have the following questions/ concerns:

– LC/MS data is very difficult to interpret/evaluate. I have questions as to: Is there any other Asn/Asp on TRAF-6 that gets modified by PIMT at all? What is the peptide coverage? What is the confidence level on distinguishing signal from noise. What is methylated Asn?- Are authors referring to L-isoaspartyl methyl ester? I am unable to identify corresponding modification by the given data.

– I'm having a hard time understanding why TRAF-6 would be stuck at L-isoaspartyl methyl ester. It is my understanding that, after PIMT reaction, succinimide formation and subsequent hydrolysis is rather spontaneous. Also, unlike TRAF-6, why are all Asns in ICAM-I modified into Asp, instead of being just "methylated?"

– Is enzymatic activity necessary for PIMT to regulate ICAM-I glycosylation? Could authors use point mutants at the active site instead of deletion (please ID how much has been deleted) to ensure proper folding and to distinguish whether PIMT binding or catalysis dominates the inhibitory effect?

– Would it be possible that ICAM-I hypo-glycosylation/attenuation simply be the downstream effect since only a small amount of PIMT is co-IP'd even (and only) when overexpressed?

– I'm a bit confused about the sequence of events and targeting mechanism of PIMT. Given that PIMT targets isoAsp, would LPS treatment (activation of TLRs) cause TRAF6 and ICAM-I to become more prone to such spontaneous modifications (how)? Can authors compare the population of modified/unmodified TRAF6 and ICAM-I under normal vs inflammatory conditions? Would the data/model here imply that isoAsp modification somehow over-activates the function of TRAF6? Wouldn't it impair the activity of TRAF6 since it would generate deformed backbone? Also, without PIMT, wouldn't the glycosylation of ICAM-I be impaired anyway since the target Asn residues would be stuck at isoAsp?

*Reviewer #2 (Recommendations for the authors):*

The authors show a role for PIMT in regulating endothelial inflammatory responses, leading to reduced NFkB activation downstream of LPS and TLR4. They demonstrate that PIMT directly interacts with TRAF6 and ICAM, modifying these targets and regulating their activity. in vivo experiments suggest a role of PIMT in regulating endothelial dysfunction and inflammation downstream of lung injury. Strengths include a thorough analysis of the interactions of PIMT with TRAF6 and ICAM and a demonstration of the modification of these targets by PIMT. Weaknesses include heavy reliance on overexpression of PIMT to observe biological effects and a weak effect of PIMT heterozygosity in vivo. This study should spur further investigation into the role of PIMT not just in repair of damaged proteins, but as a possible signalling regulator of inflammatory responses.

Recommendations

– Since the paper is heavily reliant on overexpression of PIMT, strengthening the data examining endogenous PIMT would greatly strengthen the paper. A second shRNA should be used in Figures 2C, 2F, and 5D. Showing a role for endogenous PIMT in regulating LPS-induced genes as in Figure 2D would also be helpful.

– Since some of the endogenous IPs are fairly dirty and it's not always clear which band is the correct band, it would be helpful to add a knockdown of the co-IPed protein to these IP experiments.

– Knockdown ICAM to show that the lower "hypo-glycosylated" band is specific for ICAM.

– Can the authors show an effect of endogenous PIMT on the ubiquitination of endogenous TRAF6? In Figure 4, the authors IP Flag-TRAF6 and blot for HA-ubiquitin. It would be predicted that ubiquitinated TRAF6 would be detected similarly with HA and Flag antibodies; however, there are no higher molecular weight bands in the Flag blot. Does this suggest that the HA blot is not specific to TRAF6?

– The LCMS data is not easy to understand, particularly in Figure 5. Could the authors give a better explanation of what the labeled peaks mean? Can the authors also show data from the methylated peptide in Figure 4D does not occur in shPIMT cells? Is there evidence of Asn to Asp shift at N350 due to PIMT activity? Can the authors show that Asn to Asp shift in Figure 5 does not occur in Ad-LacZ cells? Is endogenous PIMT enough for this to occur?

– Since the authors don't show an actual analysis of the glycosylation of ICAM, they should refrain from using "hypo-glycosylated" and "full-glycosylated". Their data strongly suggests that PIMT is changing the glycosylated residues of ICAM, but these terms are overstatements unless they directly measure glycosylation of ICAM.

– In Figure 5E, S3B, authors say that PIMT interacts only with hypoglycosylated ICAM. Since they IP only ICAM and their IP antibody does not distinguish from 100kd and 70kd bands, they cannot say this. They need to IP PIMT and show that only one ICAM band comes down in this IP. They should also use a dose of Ad-PIMT at which there is approximately equal higher and lower molecular weight ICAM bands.

– It is a little concerning that no endogenous PIMT comes down with ICAM pulldown in Figure S3B. Can the authors show that PIMT knockdown leads to increased surface localization of ICAM with or without TNF, and that PIMT knockdown leads to increased adhesion of THP1 cells?

*Reviewer #3 (Recommendations for the authors):*

The authors studied the regulatory role that PIMT plays in endothelial cells' response to innate stimulation. They found that PIMT modulates LPS-induced NF-κB activation by interaction with TRAF6 and methylation of Asn residues in its coiled-coil domain. Therefore, PIMT inhibits TRAF6 oligomerization and autoubiquitination. In addition, the authors found that PIMT inhibits N-glycosylation of ICAM-1 and prevents ICAM-1 expression on endothelial cell surface. In a LPS-induced acute lung injury model, the authors showed that PIMT ameliorates lung inflammation and injury. The study suggests that PIMT limits endothelial cell activation and prevents excessive tissue inflammation.

Using overexpression and knockdown methods, the authors have done elegant work in vitro to show how PIMT regulates endothelial activation. Whether it is the case in vivo needs to be tested.

There are some concerns that need to be addressed or clarified.

1. The authors focused on endothelial PIMT. It seems PIMT is expressed in many kinds of cells, such as macrophages (http://biogps.org/#goto=genereport&id=18537). The authors showed that PIMT is expressed in CD31+ endothelial cells in the lung using IHF. Do other cells in the lung also express PIMT? The authors may use FACS to measure PIMT expression in lung endothelial cells, epithelial cells and immune cells. Alternatively, an endothelial specific KO mice may be used.

2. The authors found that PIMT regulates endothelial activation and ICAM-1 expression in vitro. in vivo evidence is also needed in the acute lung injury model.

3. TRAF6 is an E3 ligase. Does PIMT regulate the polyubiquitination and phosphorylation of its substrate *TAK1*?

[Editors’ note: further revisions were suggested prior to acceptance, as described below.]

Thank you for resubmitting your work entitled "PIMT: a novel and potent suppressor of endothelial activation" for further consideration by *eLife*. Your revised article has been evaluated by Carla Rothlin (Senior Editor) and a Reviewing Editor.

The manuscript has been improved. However, we find the LC/MS data still a bit incomplete and noisy. Can authors show the degree of modification of ICAM by endogenous PIMT? Similarly, could authors show the ratio between unmodified and modified peptides by (ectopically expressed) PIMT? These data will be crucial for establishing the physiological relevance of ICAM modification by PIMT.

*Reviewer #1 (Recommendations for the authors):*

The authors have largely addressed my concerns; however several issues remain:

I remain concerned about the lack of evidence for endogenous PIMT in the regulation of ICAM1. The Western blot that the authors include for shPIMT cells shows no effect on the lower molecular weight band (cropped from Figure 5C, but included in the full image). If the authors treated those cells with TNFa, IL1B, or LPS, would the increased amount of total ICAM be sufficient to see differences in low molecular weight ICAM1? The authors show that low molecular weight ICAM1 is present on the membrane of AD-LacZ cells. Is this small amount of low molecular weight ICAM reduced by knockdown of PIMT? It's worth mentioning that the ICAM1 bands in Figures 5C and 6C don't line up with the ladder in the same way as they do in other figures. Was a different ladder used for these blots?

The LC/MS data remains difficult to evaluate. Presumably, the TRAF6 analyzed is a mixture of modified and non-modified peptides. I would assume that not every single TRAF6 in the cell contains methylated asparagines and that the ~50% knockdown of PIMT would not completely abolish all of these methylations. Is there any way to represent the data that shows the relative amount of PIMT-modified to unmodified TRAF6 present in cells? This would presumably be more difficult in ICAM given the issues with glycosylation.

The authors' change of hypo- and full- glycosylated to partially and highly glycosylated misses the point. They have not analyzed glycosylation of ICAM at all. It is possible (even highly likely) that the two different bands represent different glycosylation states of ICAM, but without experimentally demonstrating this, the authors should stick to high molecular weight and low molecular weight ICAM throughout. It is fine to suggest that the effects of PIMT are likely due to differential glycosylation, but statements like "we uncovered an unexpected role of PIMT in modulating N-glycosylation of adhesion molecules", "PIMT specially inhibits the N-glycosylation of ICAM1 and VCAM1", and "PIMT blocked ICAM1 expression by inhibiting N-glycosylation" go beyond what the data show.

*Reviewer #2 (Recommendations for the authors):*

The authors have addressed my concerns.

---

## [Author Response]

[Editors’ note: the authors resubmitted a revised version of the paper for consideration. What follows is the authors’ response to the first round of review.]

Essential revisions:.Reviewer #1 (Recommendations for the authors):The authors found that impairing PIMT (KD) exacerbated inflammation and vascular leakage upon LPS challenge. They report here that this is due to the lack of repair on damaged Asn on TRAF6, which resulted in inhibition of its oligomerization and ubiquitination (consequently preventing NF-κB transactivation). The authors also found that PIMT was linked to regulating ICAM-1 expression by inhibiting N-glycosylation. This study reveals a previously unappreciated function of PIMT in regulating innate immune responses- limiting inflammatory responses thus maintaining vascular homeostasis.

Thank you for your comments and helping us to improve the manuscript.

I have the following questions/ concerns:– LC/MS data is very difficult to interpret/evaluate. I have questions as to: Is there any other Asn/Asp on TRAF-6 that gets modified by PIMT at all?

Great question. We focused on the CC domain because our domain mapping studies demonstrated this domain to be responsible for the binding of PIMT on TRAF6. This led us to hypothesize that PIMT may modify Asn/Asp residues in this domain. Ideally, this could be confirmed by overexpressing the CC domain but the small size (60 aa) of this region makes such a strategy almost impossible. As such, we choose to overexpresss a larger TRAF6 fragment containing the entire c-terminal domain (aa: 289-522). Doing this, we uncovered by LC/MS that both the D339 and N350 residues were modified in PIMT expressing HEK293 cells, but not in PIMT knockdown cells.

Functional studies also showed that the N350 modification is critically important in the regulation of TRAF6 by PIMT. Although we suspect that PIMT may also modify Asn/Asp residues outside of the cterminal domain we feel these experiments are beyond the scope of this manuscript.

What is the peptide coverage? What is the confidence level on distinguishing signal from noise.

We have a high degree of confidence in our peptide identification. In mass spec studies, we overexpressed TRAF6 fragment in HEK293 cells and extracted immunoprecipitated TRAF6 from gels for LC/MS analysis. We then followed standard guidelines established by the proteomic research societies (https://www.mcponline.org/mass-spec-guidelines) for next steps. Our false discovery rate threshold for peptide identification was less than 1%. Moreover, since we used high-resolution mass spectrometry, the peptide mass tolerance was 10 ppm for identification. More importantly, the identification of peptides was further validated by manual inspection of MS/MS data to ensure accurate peptide and PTM site assignments. Using this approach, we identified clear fragment ions corresponding to methylated Asn in our MS/MS spectra. Altogether, we believe our approach utilized gold standard methods for performing PTM localization and have a high degree of confidence in our ability to distinguish signal from noise.

What is methylated Asn?- Are authors referring to L-isoaspartyl methyl ester? I am unable to identify corresponding modification by the given data.

Sorry for the confusion. Methylated Asn is actually L-isoaspartyl methyl ester. In the revised figure 4D, the formation of L-isoaspartyl methyl ester from Asn is now clearly depicted. Peptide mass was matched with 1.5 ppm accuracy to the theoretical mass of L-isoaspartyl methyl ester (881.4211, z=2) instead of 874.4120 for the unmodified form. Also, diagnostic fragmentation ions such as 147.08681 (N-me), 307.11740 (CN-me), 435.17598 (QCN-me) corresponding to the Asn-derived Lisoaspartyl methyl ester were observed.

– I'm having a hard time understanding why TRAF-6 would be stuck at L-isoaspartyl methyl ester. It is my understanding that, after PIMT reaction, succinimide formation and subsequent hydrolysis is rather spontaneous.

We agree that isoAsp repair is traditionally recognized as a spontaneous process. However, recent studies show that O-methylation of IsoAsp occurs in nature, including on proteins such as p53 (Nat Commun. 2012;3:927) and histone H4 (Sci Rep. 2014;4:6674), and exerts important biological functions. Notably, a recent publication demonstrated that O-methylation of p53 is not just a transient event and can be maintained for at least 24 hours (Nat Commun. 2012;3:927). We have added these comments into the Discussion section of the revised manuscript (page 26, from bottom, line 5-9).

Also, unlike TRAF-6, why are all Asns in ICAM-I modified into Asp, instead of being just "methylated?"

Very thoughtful but challenging question. Our data show that among 8 Asns that were identified to be glycosylated in ICAM-1, only 4 Asns were modified to Asp in the presence of PIMT. Although we do not have a great answer regarding the specificity, it is likely that isoAsp repair and maintenance, just like other protein modifications, occur in a context and cell type dependent fashion. We have now modified the Discussion to address this criticism (page 28, 1^st^ paragraph, line 8-11).

– Is enzymatic activity necessary for PIMT to regulate ICAM-I glycosylation? Could authors use point mutants at the active site instead of deletion (please ID how much has been deleted) to ensure proper folding and to distinguish whether PIMT binding or catalysis dominates the inhibitory effect?

Thank you for your suggestion. Yes enzymatic activity of PIMT is necessary to regulate ICAM-I glycosylation since we found that modification of 4 Asns in ICAM-I needs the interaction of PIMT with ICAM1 and a subsequent PIMT catalyzed o-methylation of IsoAsp. In our preliminary studies, we generated PIMT catalytic null mutations including G88A (Nat Commun. 2012;3:927) and D83V (Front Genet. 2021 Jan 21;11:612343) and found that they still maintain some enzymatic activity as compared with methyltransferase motif I (81-90) deleted mutant (Arch Biochem Biophys. 1994;310:417). Thus, we used an enzymatic inactive PIMT deletion mutant (deletion of aa 81-89) in our study. It seems that this small deletion (9 aa) will not significantly affect the proper protein folding, based on a previous report (Arch Biochem Biophys. 1994;310:417). Some of the comments have been added into the Discussion section of the revised manuscript (page 27, from bottom, line 5-6 ).

– Would it be possible that ICAM-I hypo-glycosylation/attenuation simply be the downstream effect since only a small amount of PIMT is co-IP'd even (and only) when overexpressed?

Our data show that PIMT associates with the low MW ICAM-1 band. However, in our IP experiment, only small fractions of PIMT co-IP’d with LMW ICAM-1. Although we suspect this could be explained by a “hit and run” mechanism, we cannot exclude the possibility that PIMT affects any one of the many steps along the pathway to protein glycosylation, such as ER quality control or glucoside attachment. In future investigations we hope to further answer these questions and have added new text to the Discussion to address this concern in the revised manuscript (page 28, 1st paragraph, line 10-14).

– I'm a bit confused about the sequence of events and targeting mechanism of PIMT. Given that PIMT targets isoAsp, would LPS treatment (activation of TLRs) cause TRAF6 and ICAM-I to become more prone to such spontaneous modifications (how)?

Yes, we believe that LPS should cause TRAF6 and ICAM-I to undergo spontaneous modifications. This is because LPS is known to trigger production of reactive oxygen species (ROS) (J Biol Chem. 2002;277(25):22131-9.), and ROS is known to induce isoAsp generation (Free Radic Biol Med. 2001 Jul 1;31(1):1-9.). This comment has been added to the Discussion section of the revised manuscript (page 26, 2^nd^ paragraph, line 8-9).

Can authors compare the population of modified/unmodified TRAF6 and ICAM-I under normal vs inflammatory conditions?

Unfortunately, measuring O-methylation of TRAF6 by LC/MS/MS is not possible at this point. Due to the fact that antibodies against O-methylation of TRAF6 do not exist, this also limits our ability to evaluate this modification using western blot or ELISA. Finally, O-methylation of ICAM-1 seems spontaneous and rapid, which does not permit its quantification by LC/MS.

Would the data/model here imply that isoAsp modification somehow over-activates the function of TRAF6? Wouldn't it impair the activity of TRAF6 since it would generate deformed backbone? Also, without PIMT, wouldn't the glycosylation of ICAM-I be impaired anyway since the target Asn residues would be stuck at isoAsp?

In addition to altering the backbone, IsoAsp modifications will likely alter conformation and post-translational modifications, which could have positive (e.g. p53) or negative (e.g. tubulin) effects on protein function. In our study, we found that PIMT knockdown enhances TRAF6/NF-κB activation (Figure 2F), suggesting that IsoAsp modifications act to enhance TRAF6 function.

Under inflammatory conditions Asn resides in ICAM are stable and resistant to deamination (Figure 5G). Thus, without PIMT we expect elevated expression and full glycosylation of ICAM. Consistent with this, our studies indicate that targeted activation of PIMT in ECs reduces ICAM-1 expression and glycosylation and mitigates endothelial activation in ALI. These proof of concept studies support using PIMT activators in the treatment of inflammatory diseases. Comments related to this have been added into the Discussion section of the revised manuscript (page 26, 2nd paragraph, line 9-13).

Reviewer #2 (Recommendations for the authors):The authors show a role for PIMT in regulating endothelial inflammatory responses, leading to reduced NFkB activation downstream of LPS and TLR4. They demonstrate that PIMT directly interacts with TRAF6 and ICAM, modifying these targets and regulating their activity. in vivo experiments suggest a role of PIMT in regulating endothelial dysfunction and inflammation downstream of lung injury. Strengths include a thorough analysis of the interactions of PIMT with TRAF6 and ICAM and a demonstration of the modification of these targets by PIMT. Weaknesses include heavy reliance on overexpression of PIMT to observe biological effects and a weak effect of PIMT heterozygosity in vivo. This study should spur further investigation into the role of PIMT not just in repair of damaged proteins, but as a possible signalling regulator of inflammatory responses.Recommendations– Since the paper is heavily reliant on overexpression of PIMT, strengthening the data examining endogenous PIMT would greatly strengthen the paper. A second shRNA should be used in Figures 2C, 2F, and 5D. Showing a role for endogenous PIMT in regulating LPS-induced genes as in Figure 2D would also be helpful.

Thank you for your suggestion. We have performed additional experiments using a second shRNA to deplete PIMT in ECs. Our new results are added into the revised figure 2C, 2F, and 5C. qPCR detection of inflammatory molecules in response to LPS in control and PIMT depleted ECs was also performed and is now shown in the revised figure 2D.

– Since some of the endogenous IPs are fairly dirty and it's not always clear which band is the correct band, it would be helpful to add a knockdown of the co-IPed protein to these IP experiments.

Thank you for your suggestion. As shown in Figure 3E, the expression of TRAF6 in ECs is relatively low, which may lead to weak interactions in IP. To fully verify that TRAF6 indeed interacts with PIMT, we performed knockdown studies using TRAF6 shRNA. As shown in Figure 3E, TRAF6 with a MW of 60 kDa in control shRNA transfected cells, but not in TRAF6 shRNA transfected cells, interacts with PIMT in ECs. This interaction is further enhanced by LPS stimulation.

– Knockdown ICAM to show that the lower "hypo-glycosylated" band is specific for ICAM.

Thank you for your comments. Although we did not perform ICAM-1 knockdown studies, we cut the lower "partially-glycosylated" band from the gel (Figure 5F) and submitted it for mass spectrometric analysis. Our results clearly identified peptide fragments of ICAM-1 (data not shown) in the band, supporting the notion that our lower band is derived from ICAM-1.

– Can the authors show an effect of endogenous PIMT on the ubiquitination of endogenous TRAF6? In Figure 4, the authors IP Flag-TRAF6 and blot for HA-ubiquitin. It would be predicted that ubiquitinated TRAF6 would be detected similarly with HA and Flag antibodies; however, there are no higher molecular weight bands in the Flag blot. Does this suggest that the HA blot is not specific to TRAF6?

As shown in many other studies, detection of endogenous ubiquitination of TRAF6 in ECs is very challenging. We speculate this relates, in large part, to the lack of high quality TRAF6 antibodies as well as the relatively low expression of TRAF6 in ECs. As shown in Figure 4A, by using protein ectopic expression, the poly-ubiquitination chain could be specifically and effectively recognized by HA antibody. Since ubiquitin is HA tagged, western blot using HA antibody only detects ubiquitinated TRAF6. After IP, western blot using Flag antibody detects the total TRAF6, which includes both ubiquitinated and non-ubiquitinated TRAF6. Since the level of ubiquitinated TRAF6 is much lower compared with non-ubiquitinated TRAF6 in the immunoprecipitates we believe this explains why we normally observe a strong total TRAF6 band in western blot using anti-Flag antibody. Through increasing exposure time, we may observe higher molecular weight bands, but the quality of image would be much lower.

– The LCMS data is not easy to understand, particularly in Figure 5. Could the authors give a better explanation of what the labeled peaks mean?

We apologize for not presenting our data clearly. We now provide a better explanation of what the labelled peaks mean in the revised Figure 5G and Figure 5-supplement 2. In Figure 5G and Figure 5-supplement 2, the mass spectrometry spectra demonstrate that DHHGANFSCR is deamidated (DHHGADFSCR). Peptide mass was matched with 1.5 ppm accuracy to the theoretical mass of the Asn-methylated form (m/z 401.1649, double charge) instead of m/z 400.8370 (double charge) for the unmodified form. In MS/MS, diagnostic fragmentation ions such as

633.3015(DHHGAD), 755.3333(DFSCR),794.31586 (ADFSCR), 812.3377 (GADFSCR) corresponding to the deamidation were observed.

Can the authors also show data from the methylated peptide in Figure 4D does not occur in shPIMT cells? Is there evidence of Asn to Asp shift at N350 due to PIMT activity? Can the authors show that Asn to Asp shift in Figure 5 does not occur in Ad-LacZ cells? Is endogenous PIMT enough for this to occur?

Unmethylated TRAF6 peptide from shPIMT cells was detected and shown in the revised Figure 4—figure supplement 1F. Asn to Asp shift, known as the repaired isoAsp, requires PIMT catalyzed O-methylation. As shown in the revised Figure 4D, O-methylation was only detected in PIMT expressing cells, but not in PIMT knockdown cells, strongly indicating that PIMT is required to methylate isoAsp derived from Asn or Asp.

In Ad-lacZ cells, ICAM-1 is fully glycosylated and after trypsin digestion, identification of N-linked glycopeptides (glycans plus peptides) using LC/MS/MS is very challenging. In our study, we only detected a peptide spectrum of ICAM-1 with an undeamidated N-glycosite in Ad-LacZ cells as shown in Author response image 1, indicating this site is neither glycosylated nor deamidated.

**Author response image 1. sa2fig1:** 

After knocking down endogenous PIMT, we observed increased expression of ICAM-1 at MW 100 kDa (Figure 5C), mainly due to a rapid induction of ICAM-1 expression by NF-ΚB activation.Proinflammatory stimuli cause a rapid upregulation in adhesion molecules and ER stress markers. In this case, some neo-synthesized ICAM1 proteins without proper folding might escape from ER quality check. Thus, we sometimes observe weak lower MW ICAM1 bands in western blot, indicating endogenous PIMT might be functional here. Nevertheless, our data show that overexpression of PIMT markedly suppresses ICAM-1 glycosylation and provides proof of concept that targeted activation of PIMT in ECs may be beneficial for inhibition of ICAM-1 function under inflammatory conditions.

– Since the authors don't show an actual analysis of the glycosylation of ICAM, they should refrain from using "hypo-glycosylated" and "full-glycosylated". Their data strongly suggests that PIMT is changing the glycosylated residues of ICAM, but these terms are overstatements unless they directly measure glycosylation of ICAM.

Thank you for your comments. We now use “partially-glycosylated” and “highly-glycosylated” instead of "hypo-glycosylated" and "full-glycosylated", respectively, in the revised manuscript.

– In Figure 5E, S3B, authors say that PIMT interacts only with hypoglycosylated ICAM. Since they IP only ICAM and their IP antibody does not distinguish from 100kd and 70kd bands, they cannot say this. They need to IP PIMT and show that only one ICAM band comes down in this IP. They should also use a dose of Ad-PIMT at which there is approximately equal higher and lower molecular weight ICAM bands.

Thank you for the suggestion. We have now performed IP using anti-PIMT antibody and results are shown in the revised Figure 5E. Although results show an equal input of high and low molecular weight ICAM bands only partially glycosylated ICAM-1 interacted with PIMT.

– It is a little concerning that no endogenous PIMT comes down with ICAM pulldown in Figure S3B.

We understand your concern; however, our data indicate that PIMT mostly associates with partially-glycosylated ICAM-1. In Ad-LacZ cells, since ICAM-1 is known to be highly glycosylated, we are unable to see endogenous PIMT1 in ICAM-1 immunocomplexes (highly glycosylated). Hopefully this explanation alleviates some of this reviewer’s concerns.

Can the authors show that PIMT knockdown leads to increased surface localization of ICAM with or without TNF, and that PIMT knockdown leads to increased adhesion of THP1 cells?

As suggested, we have performed PIMT knockdown and examined the surface localization of ICAM by flow cytometry. As shown in the revised figure 5—figure supplement 1C, we found that PIMT knockdown significantly increased cell surface ICAM-1 expression. Consistent with this, we found that PIMT knockdown also increased the adhesion of THP1 cells to activated ECs. These data are now shown in the revised figure 5—figure supplement 1D.

Reviewer #3 (Recommendations for the authors):The authors studied the regulatory role that PIMT plays in endothelial cells' response to innate stimulation. They found that PIMT modulates LPS-induced NF-κB activation by interaction with TRAF6 and methylation of Asn residues in its coiled-coil domain. Therefore, PIMT inhibits TRAF6 oligomerization and autoubiquitination. In addition, the authors found that PIMT inhibits N-glycosylation of ICAM-1 and prevents ICAM-1 expression on endothelial cell surface. In a LPS-induced acute lung injury model, the authors showed that PIMT ameliorates lung inflammation and injury. The study suggests that PIMT limits endothelial cell activation and prevents excessive tissue inflammation.Using overexpression and knockdown methods, the authors have done elegant work in vitro to show how PIMT regulates endothelial activation. Whether it is the case in vivo needs to be tested.

Thank you for your comments and helping us to improve the manuscript.

There are some concerns that need to be addressed or clarified.1. The authors focused on endothelial PIMT. It seems PIMT is expressed in many kinds of cells, such as macrophages (http://biogps.org/#goto=genereport&id=18537). The authors showed that PIMT is expressed in CD31+ endothelial cells in the lung using IHF. Do other cells in the lung also express PIMT? The authors may use FACS to measure PIMT expression in lung endothelial cells, epithelial cells and immune cells. Alternatively, an endothelial specific KO mice may be used.

Thank you for your suggestion. Yes, PIMT is also expressed in macrophages and epithelial cells in the lung. In this study, we did not perform FACS to determine the relative expression of PIMT in different types of cells in the lung. Data documented in the public domain of The Human Protein Atlas (https://www.proteinatlas.org/ENSG00000120265-PCMT1/tissue/lung) provides a single cell transcriptome analysis in human lung tissues demonstrating that PIMT is highly expressed in lung endothelial cells, as shown in Author response image 2. Although we did not use endothelial specific KO mice, our in vitro experiments using gain- and loss-of function strategies clearly demonstrated that PIMT is important for lung endothelial activation. Determination of relative contribution of PIMT in different cell types to ALI requires generation of various tissue specific knockout mice, which is extremely time-consuming, will need to be the subject of future studies. The comments have been added into the Discussion section of the revised manuscript (page 27, 1^st^ paragraph, line 3-6).

2. The authors found that PIMT regulates endothelial activation and ICAM-1 expression in vitro. in vivo evidence is also needed in the acute lung injury model.

Thank you for your suggestion. As shown in Figure 1B and E, we now demonstrate that neutrophil infiltration and ICAM-1 expression (both are hallmarks of endothelial activation) are increased in the lungs of PIMT knockout mice after LPS when compared to wild-type littermates.

3. TRAF6 is an E3 ligase. Does PIMT regulate the polyubiquitination and phosphorylation of its substrate TAK1?

Excellent point. We have now performed an ubiquitination assay and found that *TAK1* polyubiquitination was effectively inhibited by active PIMT, but not by its catalytic null mutant.

Furthermore, we found that *TAK1* phosphorylation (S412) was negatively regulated by PIMT in ECs. These results have now been incorporated into the revised figure 4—figure supplement 1A and 1B.

[Editors’ note: what follows is the authors’ response to the second round of review.]

Reviewer #1 (Recommendations for the authors):The authors have largely addressed my concerns; however several issues remain:I remain concerned about the lack of evidence for endogenous PIMT in the regulation of ICAM1. The Western blot that the authors include for shPIMT cells shows no effect on the lower molecular weight band (cropped from Figure 5C, but included in the full image). If the authors treated those cells with TNFa, IL1B, or LPS, would the increased amount of total ICAM be sufficient to see differences in low molecular weight ICAM1? The authors show that low molecular weight ICAM1 is present on the membrane of AD-LacZ cells. Is this small amount of low molecular weight ICAM reduced by knockdown of PIMT? It's worth mentioning that the ICAM1 bands in Figures 5C and 6C don't line up with the ladder in the same way as they do in other figures. Was a different ladder used for these blots?

Thank you for the comments. Our results indicate that PIMT regulates ICAM-1 expression by affecting both NF-ΚB activation (transcriptional mechanism) and modifying protein glycosylation (posttranslational mechanism). Our results indicate that transcriptional mechanisms are more significant in PIMT knockdown cells since this mainly causes an increase in NF-ΚB activation without affecting N-linked glycosylation of ICAM-1. Consistent with this, we did not observe a change in the 8 ICAM-1 glycosylated sites (Asn) in shPIMT cells and we would not expect to find changes in the lower molecular weight band in Figure 5C. We also anticipate that treatment of shPIMT cells with other NF-ΚB activators, such as TNFa, IL1B, or LPS, would not significantly alter the amount of low molecular weight ICAM on gels because ICAM-1 should be highly glycosylated in this setting. However, this needs to be confirmed in future studies.

As shown in Figure 6C, the size of low molecular weight (LMW) ICAM-1 is different on the membrane of AD-LacZ and Ad-PIMT cells. Although we do not have an explanation for these findings we speculate this relates to different types of N-glycosylation or to other types of protein modifications. Further, our data indicate that ICAM-1 is highly glycosylated in PIMT knockdown cells. Based on this, we speculate that smaller amounts of LMW ICAM-1 present on the membrane of AD-LacZ cells will be shifted to HMW ICAM-1 in PIMT knockdown cells.

Yes, we used 2 types of protein ladders in this study, which were described in the Key Resources Table.

The LC/MS data remains difficult to evaluate. Presumably, the TRAF6 analyzed is a mixture of modified and non-modified peptides. I would assume that not every single TRAF6 in the cell contains methylated asparagines and that the ~50% knockdown of PIMT would not completely abolish all of these methylations. Is there any way to represent the data that shows the relative amount of PIMT-modified to unmodified TRAF6 present in cells? This would presumably be more difficult in ICAM given the issues with glycosylation.

We appreciate the valuable advice provided and completely agree with your comments. In order to investigate the degree of modification, we revisited the mass spectrometry spectra. We found that two forms of protein co-eluted in our HPLC and that both forms had a similar intensity in our analysis. As a result, the comment that "the ~50% knockdown of PIMT would not completely abolish all of these methylations " is absolutely correct. The relative amount of modified to unmodified protein at N350 in TRAF6-CC domain is now presented in Figure 4—figure supplement 1F and the revised manuscript (page 21, 2^nd^ paragraph, line 9-10).

Indeed, it is more difficult to show the modification of ICAM-1 in PIMT knockdown cells as a significant amount of ICAM-1 will be highly glycosylated.

The authors' change of hypo- and full- glycosylated to partially and highly glycosylated misses the point. They have not analyzed glycosylation of ICAM at all. It is possible (even highly likely) that the two different bands represent different glycosylation states of ICAM, but without experimentally demonstrating this, the authors should stick to high molecular weight and low molecular weight ICAM throughout. It is fine to suggest that the effects of PIMT are likely due to differential glycosylation, but statements like "we uncovered an unexpected role of PIMT in modulating N-glycosylation of adhesion molecules", "PIMT specially inhibits the N-glycosylation of ICAM1 and VCAM1", and "PIMT blocked ICAM1 expression by inhibiting N-glycosylation" go beyond what the data show.

As suggested, we now use the terms high molecular weight (HMW) and low molecular weight (LMW) to refer to different forms of ICAM-1 and have modified other statements in the manuscript to more clearly represent our data.